# Occurrence of foamy macrophages during the innate response of zebrafish to trypanosome infections

Sem H Jacobs[1,2], Eva Dóró[1†], Ffion R Hammond[1‡], Mai E Nguyen-Chi[3], Georges Lutfalla[3], Geert F Wiegertjes[1,4], Maria Forlenza[1]*

[1]Cell Biology and Immunology Group, Department of Animal Sciences, Wageningen University & Research, Wageningen, Netherlands; [2]Experimental Zoology Group, Department of Animal Sciences, Wageningen University & Research, Wageningen, Netherlands; [3]DIMNP, CNRS, University of Montpellier, Montpellier, France; [4]Aquaculture and Fisheries Group, Department of Animal Sciences, Wageningen University & Research, Wageningen, Netherlands

**Abstract** A tightly regulated innate immune response to trypanosome infections is critical to strike a balance between parasite control and inflammation-associated pathology. In this study, we make use of the recently established *Trypanosoma carassii* infection model in larval zebrafish to study the early response of macrophages and neutrophils to trypanosome infections in vivo. We consistently identified high- and low-infected individuals and were able to simultaneously characterise their differential innate response. Not only did macrophage and neutrophil number and distribution differ between the two groups, but also macrophage morphology and activation state. Exclusive to high-infected zebrafish, was the occurrence of foamy macrophages characterised by a strong pro-inflammatory profile and potentially associated with an exacerbated immune response as well as susceptibility to the infection. To our knowledge, this is the first report of the occurrence of foamy macrophages during an extracellular trypanosome infection.

*For correspondence:
maria.forlenza@wur.nl

Present address: †Institute of Physiology, Faculty of Medicine, University of Pécs, Pécs, Hungary; ‡ Department of Infection, Immunity & Cardiovascular Disease, University of Sheffield, Sheffield, United Kingdom

Competing interests: The authors declare that no competing interests exist.

## Introduction

Trypanosomes of the *Trypanosoma* genus are protozoan haemoflagellates that can infect animals from all vertebrate classes, including warm-blooded mammals and birds as well as cold-blooded amphibians, reptiles, and fish. This genus contains human and animal pathogens, including the intracellular *Trypanosoma cruzi* (causing Human American Trypanosomiasis or Chagas' disease), the extracellular *T. brucei rhodesiense* and *T. brucei gambiense* (causing Human African Trypanosomiasis or Sleeping Sickness) and *T. congolense*, *T. vivax* and *T. b. brucei* (causing Animal African Trypanosomiasis or Nagana) (*Radwanska et al., 2018*; *Simpson et al., 2006*). Among these, salivarian trypanosomes such as *T. brucei* ssp. live extracellularly in the bloodstream or tissue fluids of their host. For example, *T. vivax* can multiply rapidly and is evenly distributed throughout the cardiovascular system, *T. congolense* tends to aggregate in small blood vessels, whereas *T. brucei* especially can extravasate and multiply in interstitial tissues (reviewed by *Magez and Caljon, 2011*). Pathologically, anaemia appears to be a factor common to infections with most if not all trypanosomes although with different underlying causative mechanisms. These include, among others, erythrophagocytosis by macrophages (*Cnops et al., 2015*; *Guegan et al., 2013*), hemodilution (*Naessens, 2006*), erythrolysis through intermembrane transfer of variant surface glycoprotein (VSG) from trypanosomes to erythrocytes (*Rifkin and Landsberger, 1990*), oxidative stress from free radicals (*Mishra et al., 2017*) and mechanical damage through direct interaction of trypanosomes with erythrocytes surface (*Boada-Sucre et al., 2016*).

Immunologically, infections with trypanosomes are often associated with dysfunction and pathology related to exacerbated innate and adaptive immune responses (reviewed by *Radwanska et al., 2018*; *Stijlemans et al., 2016*). Initially, it was believed that antibody-dependent complement-mediated lysis was the major protective mechanism involved in early parasite control (*Krettli et al., 1979*; *Musoke and Barbet, 1977*). However, later studies revealed that at low antibody levels, trypanosomes can efficiently remove surface-bound antibodies through an endocytosis-mediated mechanisms (*Engstler et al., 2007*), and that complement C5-deficient mice are able to control the first-peak parasitaemia similarly to wild-type mice (*La Greca et al., 2014*). Instead, innate immune mediators such as IFNγ, TNFα, and nitric oxide (NO) were shown to be indispensable for the control of first-peak parasitaemia, through direct and indirect mechanisms (reviewed by *Radwanska et al., 2018*). In the early phase of infection, the timely induction of IFNγ by NK, NKT, and CD8[+] cells *Cnops et al., 2015* followed by the production of TNFα and NO by IFNγ-primed macrophages (*Baral et al., 2007*; *Iraqi et al., 2001*; *Lopez et al., 2008*; *Lucas et al., 1994*; *Magez et al., 1993*; *Magez et al., 2007*; *Magez et al., 2006*; *Magez et al., 2001*; *Magez et al., 1999*; *O'Gorman et al., 2006*; *Sternberg and Mabbott, 1996*; *Wu et al., 2017*) leads to effective control of first-peak parasitaemia. Glycosyl-inositol-phosphate soluble variant surface glycoproteins (GPI-VSG) released from the surface of trypanosomes were found to be the major inducers of TNFα in macrophages, and that such response could be primed by IFNγ (*Coller et al., 2003*; *Magez et al., 2002*). When macrophages would encounter GPI-VSG prior to IFNγ exposure, however, their TNFα and NO response would dramatically be reduced (*Coller et al., 2003*) which, depending on the timing, could either lead to macrophage unresponsiveness or prevent exacerbated inflammatory responses during the first-peak of parasite clearance. Altogether, these data made clear that an early innate immune response is crucial to control the acute phase of trypanosome infection, but that its tight regulation is critical to ensure parasite control as opposed to pathology.

All the findings above took advantage of the availability of several mice models for trypanosome infection using trypanosusceptible or trypanotolerant as well as mutant 'knock-out' mice strains. Although mice cannot be considered natural hosts of trypanosomes and do not always recapitulate all features of natural infections, the availability of such models allowed to gain insights into the general biology of trypanosomes, their interaction with and evasion of the host immune system, as well as into various aspects related to vaccine failure, antigenic variation, and (uncontrolled) inflammation (*Magez and Caljon, 2011*). The use of knock-out strains for example, shed specific light on the role of various cytokines, particularly TNFα, IFNγ, and IL-10, in the control of parasitaemia and in the induction of pathological conditions during infection (reviewed in *Magez and Caljon, 2011*). It would be ideal to be able to follow, in vivo, the early host responses to the infection and visualise the trypanosome response to the host's attack. However, due to the lack of transparency of most mammalian hosts, this has not yet been feasible.

We recently reported the establishment of an experimental trypanosome infection of zebrafish (*Danio rerio*) with the fish-specific trypanosome *Trypanosoma carassii* (*Dóró et al., 2019*). In the latter study, by combining *T. carassii* infection of transparent zebrafish with high-resolution high-speed microscopy, we were able to describe in detail the swimming behaviour of trypanosomes in vivo, in the natural environment of blood and tissues of a live vertebrate host. This led to the discovery of novel attachment mechanisms as well as trypanosome swimming behaviours that otherwise would not have been observed in vitro (*Dóró et al., 2019*). Previous studies in common carp (*Cyprinus carpio*), goldfish (*Carrassius aurata*) and more recently zebrafish, demonstrated that infections with *T. carassii* present many of the pathological features observed during human or animal trypanosomiasis, including a pro-inflammatory response during first-peak parasitaemia (*Kovacevic et al., 2015*; *Oladiran et al., 2011*; *Oladiran and Belosevic, 2009*) polyclonal B and T cell activation (*Joerink et al., 2007*; *Joerink et al., 2004*; *Lischke et al., 2000*; *Ribeiro et al., 2010*; *Woo and Ardelli, 2014*) and anaemia (*Dóró et al., 2019*; *Islam and Woo, 1991*; *McAllister et al., 2019*). These shared features among human and animal (including fish) trypanosomiases suggest a commonality in (innate) immune responses to trypanosomes across different vertebrates.

Zebrafish are fresh water cyprinid fish closely related to many of the natural hosts of *T. carassii* (*Kent et al., 1993*; *Simpson et al., 2006*) and are a powerful model species owing to, among others, their genetic tractability, large number of transgenic lines marking several immune cell types, knock-out mutant lines and most importantly, the transparency of developing embryos allowing high-resolution in vivo visualisation of cell behaviour (*Benard et al., 2015*; *Bertrand et al., 2010*; *Ellett et al.,*

*2011*; *Langenau et al., 2004*; *Lawson and Weinstein, 2002*; *Page et al., 2013*; *Petrie-Hanson et al., 2009*; *Renshaw et al., 2006*; *White et al., 2008*). During the first 2–3 weeks of development, zebrafish are devoid of mature T and B lymphocytes and thus offer a window of opportunity to study innate immune responses (*Torraca et al., 2014*; *Torraca and Mostowy, 2018*), especially those driven by neutrophils and macrophages. The response of macrophages and neutrophilic granulocytes towards several viral, fungal and bacterial pathogens has been studied in detail using zebrafish (*Cronan and Tobin, 2014*; *García-Valtanen et al., 2017*; *Nguyen-Chi et al., 2014*; *Palha et al., 2013*; *Ramakrishnan, 2013*; *Renshaw and Trede, 2012*; *Rosowski et al., 2018*; *Torraca and Mostowy, 2018*) but never before in the context of trypanosome infections.

Taking advantage of the recently established zebrafish-*T. carassii* infection model and of the availability of zebrafish transgenic lines marking macrophages and neutrophils as well as *il1b*- and *tnfa*-expressing cells, in the current study, we describe the early events of the innate immune response of zebrafish to *T. carassii* infections. Based on a novel clinical scoring system relying, amongst other criteria, on in vivo real-time monitoring of parasitaemia, we could consistently segregate larvae in high- and low-infected individuals without having to sacrifice the larvae. Between these individuals we always observed a marked differential response between macrophages and neutrophils, especially with respect to their proliferative capacity and redistribution in tissues or major blood vessels during infection. Significant differences were observed in the inflammatory response of macrophages in high- and low-infected individuals and in their susceptibility to the infection. In low-infected individuals, despite an early increase in macrophage number, a mild inflammatory response strongly associated with control of parasitaemia and survival to the infection was observed. Conversely, exclusively in high-infected individuals, we describe the occurrence of large, granular macrophages, reminiscent of foamy macrophages (*Vallochi et al., 2018*), characterised by a strong inflammatory profile and association to susceptibility to the infection. This is the first report of the occurrence of foamy macrophages during an extracellular trypanosome infection.

## Results

### Susceptibility of zebrafish larvae to *T. carassii* infection

We recently reported the establishment of a trypanosome infection in zebrafish larvae using a natural fish parasite, *Trypanosoma carassii* (*Dóró et al., 2019*). To further investigate the immune response to *T. carassii* infection, we first investigated the kinetics of susceptibility of zebrafish larvae as well as the kinetics of expression of various immune-related genes. Similar to the previous report, *T. carassii* infection of 5 dpf zebrafish larvae leads to approximately 10–20% survival by 15 days post infection (dpi) with the highest incidence of mortality between 4 and 7dpi (*Figure 1A*). The onset of mortality coincided with the peak of parasitaemia as assessed by real-time quantitative gene expression analysis of a *T. carassii*-specific gene (*Figure 1B*). Nevertheless, we consistently observed 10–20% survival in the *T. carassii*-infected group, suggesting that zebrafish larvae can control *T. carassii* infection. This observation prompted us to investigate the kinetics of parasitaemia and development of (innate) immune responses at the individual level.

### Clinical signs of *T. carassii* infection and clinical scoring system

To characterise the response to *T. carassii* infection in individual zebrafish larvae, we developed a clinical scoring system to determine individual infection levels, enabling us to group individual larvae based on severity of infection. From 4 dpi onwards, we could consistently sort larvae into groups of high- or low-infected individuals based on in vivo observations, without the need to sacrifice animals (*Video 1*). Infection levels were categorised using four criteria: (1) escape reflex (slow vs fast) upon contact with a pipette tip, (2) infection scores (1–10, see details in Materials and methods), based on the ratio of blood cells and parasites passing through an intersegmental capillary (ISC) in 100 events (*Figure 2A,B*) (*Video 1*, 00:06-00:39 s), (3) extravasation, based on the presence of parasites outside of blood vessels (*Figure 2C*) (*Video 1*, 00:40-1:20 s) and (4) vasodilation, based on the diameter of the caudal vein (*Figure 2D,E*). The first criterion defined all individuals with a minimal escape reflex (slow swimmers) as high-infected individuals: they were mostly located at the bottom of the tank and showed minimal reaction upon direct contact with a pipette. Larvae with a normal escape reflex (fast swimmers), however, were not exclusively low-infected individuals. Therefore, a second criterion

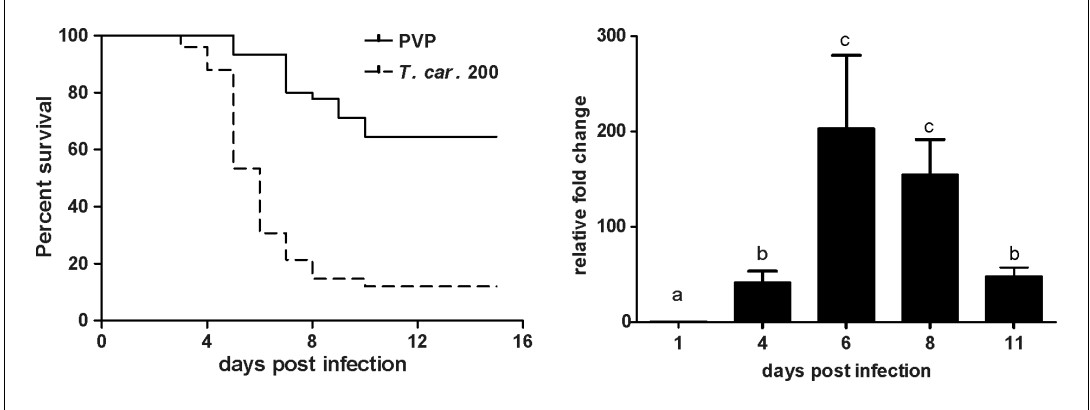

**Figure 1.** *T. carassii* infection of larval zebrafish. (**A**) *Tg(mpeg1:mCherry-F;mpx:GFP)* larvae (5 dpf) were injected intravenously with *n* = 200 *T. carassii*/fish or with PVP as control and survival was monitored over a period of 15 days. (**B**) *Tg(mpeg1:mCherry-F;mpx:GFP)* zebrafish (5 dpf) were treated as in A and sampled at various time points. At each time point, three pools of 3–5 larvae were sampled for real-time quantitative PCR analysis. Relative fold change of the *T. carassii*-specific *heat-shock protein-70* (*hsp70*) was normalised to the zebrafish-specific *ef1α* and expressed relative to the trypanosome-injected group at time point zero. Bars indicate average and standard deviation (SD) on *n* = 3 pools per time point. Letters indicate significant differences (p<0.05), as assessed using One-way ANOVA followed by Tukey's multiple comparisons test.

The online version of this article includes the following source data for figure 1:

**Source data 1.** *T. carassii* infection of larval zebrafish.

was used based on trypanosome counting in ISC (*Video 1*, 00:06-00:39 s). Individuals with an infection score of 1 (no parasites) were never observed, indicating that larvae cannot clear the infection, at least not within 4 days. Individuals with an infection score between 2 and 3 (~80%, *Figure 2F*) were categorised as low-infected and had a high survival rate (relative percent survival, 82%; *Figure 2G*). Individuals with an infection score between 6 and 10 (~20%, *Figure 2F*) were categorised as high-infected and generally succumbed to the infection (*Figure 2G*). Individuals with an intermediate score of 4–5 (~5%, *Figure 2F*) were re-evaluated at 5 dpi and could go both ways: they either showed a delayed parasitaemia and later developed high parasitaemia (common) or recovered from the infection (rare). The third criterion clearly identified high-infected individuals as those showing extensive extravasation at two or more of the following locations: peritoneal cavity (*Figure 2C*) (*Video 1*, 00, 00:40-00:59 s), interstitial space lining the blood vessels, muscle tissue (*Video 1*, 00:00-01:07 s) or fins (*Video 1*, 01:08-01:20 s), in particular the anal fin. At these locations,

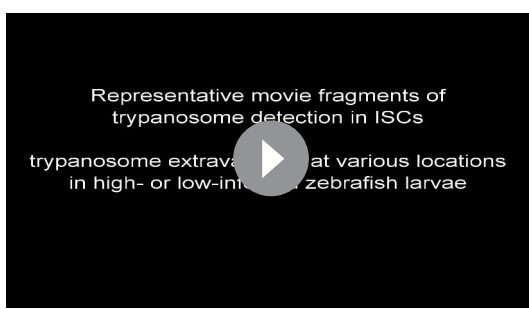

**Video 1.** Clinical signs of *T. carassii* infection in high- and low-infected zebrafish larvae. *Tg(mpeg1:mCherry-F;mpx:GFP)* 5 dpf zebrafish were injected with *n* = 200 *T. carassii* or with PVP and imaged at various time points after infection. Shown are high-speed videos (500 frames/s, fps) or real-time videos (20 fps) capturing trypanosomes in vivo in blood or in tissues, as well as describing typical signs of anaemia and vasodilation.
https://elifesciences.org/articles/64520#video1

in high-infected individuals, trypanosomes could accumulate in high numbers, filling up all available spaces. Extravasation however could also occur in low-infected individuals, but to a lesser extent. The fourth criterion, vasodilation of the caudal vein associated with high numbers of trypanosomes in the blood vessels, was a definitive sign of high infection level, and never occurred in low-infected larvae. To validate our scoring system, expression of a *T. carassii*-specific gene was analysed in pools of larvae classified as high- or low-infected. As expected, in individuals categorised as high-infected, *T. carassii*-specific gene expression increased more than 60-fold, whereas in low-infected individuals the increase was less than 20-fold (*Figure 2H*). Altogether these data show that *T. carassii* infects zebrafish larvae, but that the infection can develop differently among individuals, leading to different outcomes. The clinical scoring system based on numerous

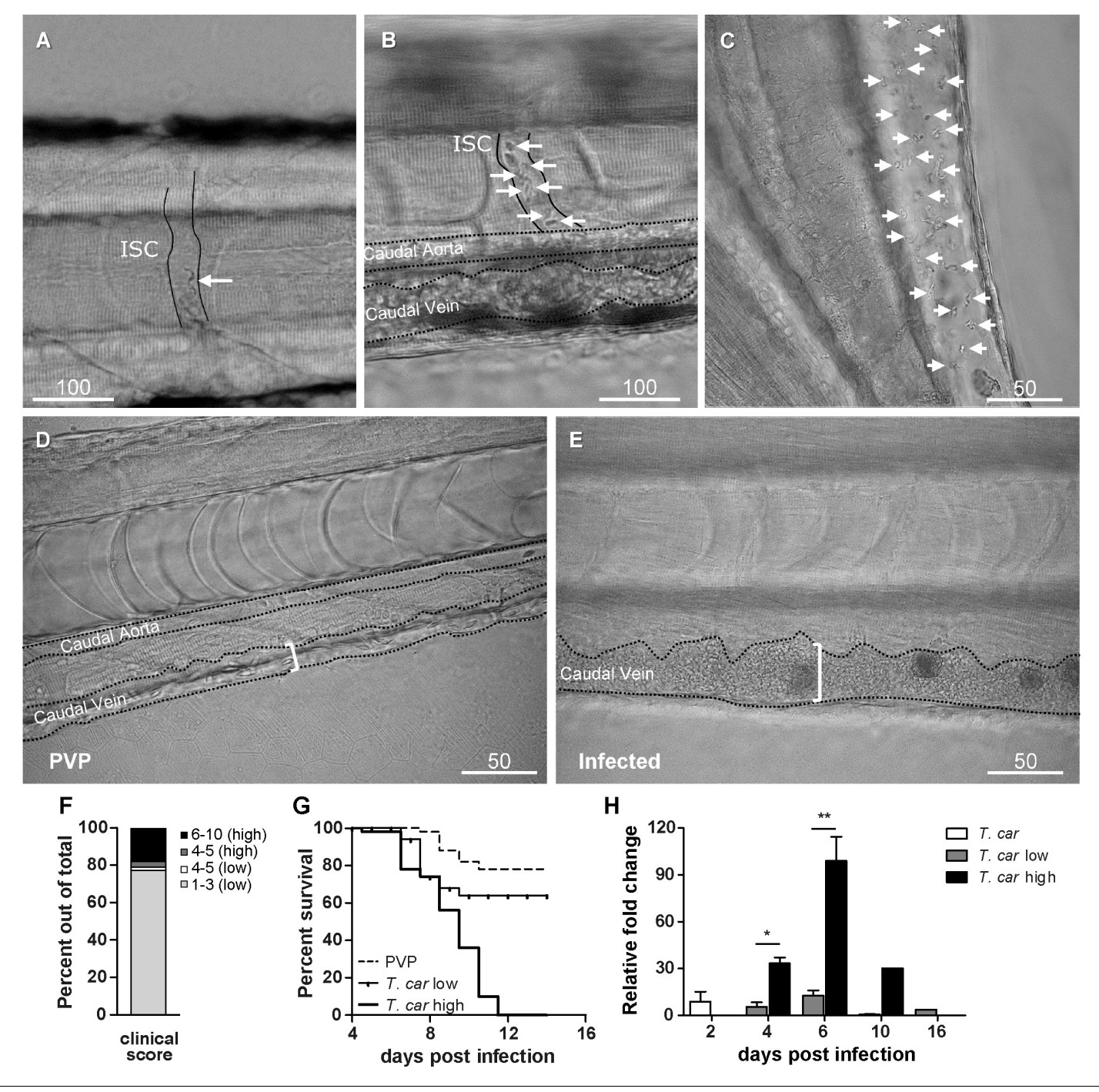

**Figure 2.** Progression of *T.carassii* infection in zebrafish larvae. *Tg(mpeg1:mCherry-F;mpx:GFP)* 5 dpf zebrafish were injected intravenously with *n* = 200 *T. carassii* or with PVP and imaged at 2 dpi (**A**), 5 dpi (**B-C**), 7 dpi (**D-E**). Shown are representative images of intersegmental capillaries (ISC) containing various numbers of *T. carassii* (white arrows) (**A-B**); extravasated *T. carassii* (only some indicated with white arrows) in the intraperitoneal cavity (**C**); caudal vein diameter in PVP (**D**) or in *T. carassii*-infected larvae (**E**). Square brackets indicate the diameter of the caudal vein. Whenever visible, the caudal aorta is also indicated. Images are extracted from high-speed videos acquired with a Leica DMi8 inverted microscope at a ×40 magnification. (**F**) *Tg(mpeg1:mCherry-F;mpx:GFP)* were injected intravenously at 5 dpf with *n* = 200 *T. carassii* and at 4 dpi the number of low-infected (clinical scores 1–3) or high-infected (score 6–10) was determined. Larvae scored between 4 and 5 were re-evaluated at 5 dpi. The bar indicates the proportion of larvae assigned to each group out of a total of 350 infected individual. (**G**) *Tg(mpeg1:mCherry-F;mpx:GFP)* were injected intravenously at 5 dpf with *n* = 200 *T. carassii* or with PVP. At 4 dpi, larvae were separated in high- and low-infected individuals (50 larvae per group) based on our clinical scoring criteria and survival was monitored over a period of 14 days. (**H**) *Tg(mpeg1:mCherry-F;mpx:GFP)* were treated as described in (**G**). At each time point, three pools

*Figure 2 continued on next page*

*Figure 2 continued*

of 3–5 larvae were sampled for subsequent real-time quantitative gene expression analysis. Each data point represents the mean of three pools, except for the low-infected group at 16 dpi and high-infected group at 10 dpi where only two and one pool could be made, respectively. Relative fold change of the *T. carassii*-specific *hsp70* was normalised relative to the zebrafish-specific *ef1α* housekeeping gene and expressed relative to the trypanosome-injected group at time point zero.

The online version of this article includes the following source data and figure supplement(s) for figure 2:

**Source data 1.** Progression of *T. carassii* infection in zebrafish larvae.
**Figure supplement 1.** Differential gene expression in high- and low-infected individuals.
**Figure supplement 1—source data 1.** Differential gene expression in high- and low-infected individuals.

criteria is suitable to reliably separate high- and low-infected larvae to further investigate individual immune responses. A preliminary gene expression analysis of a panel of immune-related genes was performed on pools of larvae classified as high- or low-infected according to our clinical scoring system. Analysis revealed a general trend for higher pro-inflammatory genes expression, including *il1b*, *tnfb*, and *il6*, in the high-infected group, but due to the large variation between pools, the differences were not significant (*Figure 2—figure supplement 1*). Furthermore, it has to be considered that the analysis was performed on pools of whole larvae, which may have obscured tissue- or cell-specific responses. For these reasons, taking advantage of the transparency of zebrafish larvae and of the established clinical scoring system, subsequent characterisation of the inflammatory response to *T. carassii* infection, was performed on individual larvae, focusing on innate immune cells.

## *T. carassii* infection induces a strong macrophage response in zebrafish larvae

After having established a method to determine infection levels in each larva, we next investigated whether a differential innate immune response would be mounted in high- and low-infected fish. To this end, using double-transgenic *Tg(mpeg1:mCherry-F;mpx:GFP)* zebrafish, we first analysed macrophage and neutrophil responses in whole larvae by quantifying total cell fluorescence in high- and low-infected individuals (*Figure 3*). Total neutrophil response (total green fluorescence) was not significantly affected by the infection (*Figure 3A and C*). In contrast, the macrophage response (total red fluorescence) increased significantly in infected individuals from 3 dpi onwards, and was most prominent in the head region and along the posterior cardinal vein and caudal vein (*Figure 3B*). In low-infected larvae, a significant increase in red fluorescence was observed already by 5 dpi and remained high up until 9 dpi; in high-infected larvae, despite a marginal but not significant increase at 5 and 7 dpi, significant differences to the PVP group were observed at day nine after infection (*Figure 3C*). Interestingly, no significant differences were observed between high- or low-infected individuals, suggesting that despite the clear differences in trypanosome levels (*Figure 2H*), overall macrophages number appeared to be influenced more by the presence than by the total number of trypanosomes.

## *T. carassii* infection leads to an increase in number of macrophages and neutrophils

The increase in overall red fluorescence can be indicative of activation of the *mpeg1* promotor driving mCherry expression, but also of macrophage proliferation. To address the latter hypothesis, *Tg(mpeg1:eGFP)* or *Tg(mpx:GFP)* zebrafish larvae were infected with *T. carassii*, and subsequently injected with iCLICK EdU for identification of dividing cells. With respect to proliferation, developing larvae display a generalised high rate of cell division throughout the body that increases overtime particularly in hematopoietic organs such as the thymus or the head kidney. Thus, for a more sensitive quantification of the response of macrophages and neutrophils to the infection, EdU was injected at 3 dpi (8 dpf), and at 4 dpi, larvae were separated in high- and low-infected individuals, followed by fixation and whole mount immunohistochemistry 6–8 hr later (30–32 hr after EdU injection). This allowed evaluating the number of dividing macrophage and neutrophil right at the onset of the macrophage response observed in (*Figure 3C*) and concomitantly with the development of differences in parasitaemia. As expected, EdU$^+$ nuclei could be identified throughout the body of developing larvae. When specifically looking at EdU$^+$ macrophage (*Figure 4*) and neutrophils

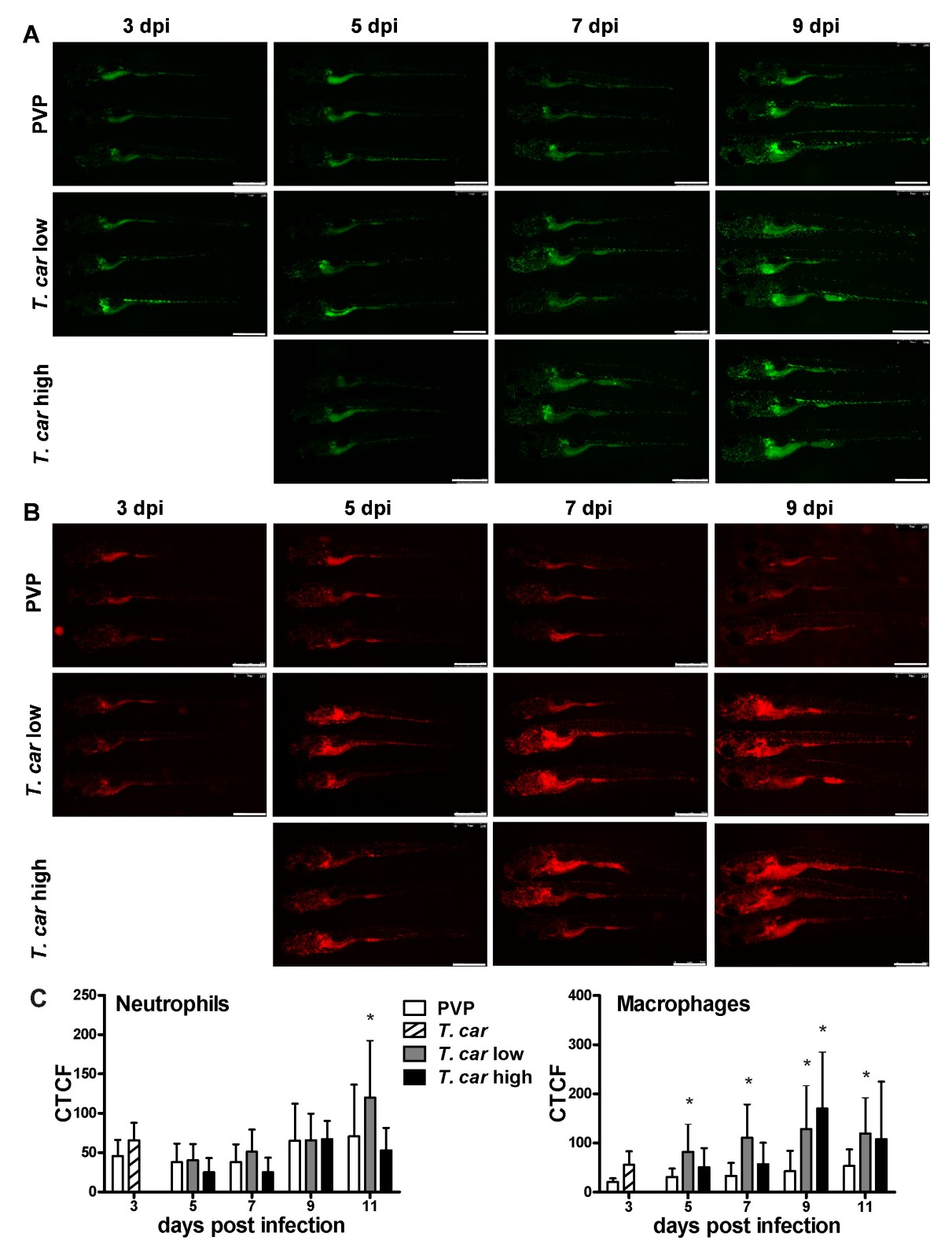

**Figure 3.** Macrophages respond more prominently than neutrophils to *T. carassii* infection. *Tg(mpeg1:mCherry-F;mpx:GFP)* were injected intravenously at 5 dpf with *n* = 200 *T. carassii* or with PVP. At 4 dpi, larvae were separated in high- and low-infected individuals. (A–B) At the indicated time points, images were acquired with Leica M205FA Fluorescence Stereo Microscope with 1.79x zoom. Images are representatives of n = 5–47 larvae per group, depending on the number of high- or low-infected larvae categorised at each time point, derived from two independent experiments. Scale bar

*Figure 3 continued on next page*

*Figure 3 continued*

indicates 750 µm. (C) Corrected Total Cell Fluorescence (CTCF) quantification of infected and non-infected larvae. Owing to the high auto-fluorescence, the gut area was excluded from the total fluorescence signal as described in the Materials and methods section. Bars represent average and standard deviation of red and green fluorescence in n = 5–47 whole larvae, from two independent experiments. * indicates significant differences (p<0.05) to the respective PVP control as assessed by Two-Way ANOVA followed by Bonferroni post-hoc test.

The online version of this article includes the following source data for figure 3:

**Source data 1.** Macrophages respond more prominently than neutrophils to *T. carassii* infection.

(*Figure 5*) we selected the area of the head (left panels) and trunk (right panels) region, where previously (*Figure 3B*) the highest increase in red fluorescence was observed.

When analysing the macrophage response, a greater number of macrophages was observed in the head and trunk of both high- and low-infected larvae compared to PVP-injected individuals (*Figures 4A*, ×10 magnifications). In the head, macrophages were scattered throughout the region but in infected larvae they were most abundant in the area corresponding to the haematopoietic tissue (head kidney), posterior to the branchial arches, indicative of proliferation of progenitor cells. In the trunk, macrophages were scattered throughout the tissue, and in high-infected larvae in particular, macrophages generally clustered in the posterior cardinal vein and caudal vein (*Figure 4A*, right panels). In agreement with previous observations (*Figure 3C*), quantification of total green fluorescence confirmed a significant increase in the head and trunk of low-infected larvae (*Figure 4B–C*). In high-infected individuals, a significant increase was observed in the trunk (*Figure 4C*), whereas in the head the number of macrophages was clearly elevated although not significantly when compared to the PVP-injected controls (*Figure 4B*). In all groups, total cell fluorescence in the head region was higher than in the trunk region (*Figure 4B–C*), and thus largely contributed to the total cell fluorescence previously measured in whole larvae (*Figure 3C*). The difference in CTCF values between *Figure 3* and *Figure 4* can be attributed to the different microscopes and magnification used for the acquisition as well as fluorescence source (GFP or mCherry in *Figure 3* and Alexa-488 fluorophore in *Figure 4*). Given the high number of macrophages in the head region, their heterogeneous morphology, the thickness of the tissue and the overall high number of EdU$^+$ nuclei, it was not possible to reliably count single (EdU$^+$) macrophages in this area. Therefore, when quantifying the number of EdU + cells, we focused on the trunk region only. There, EdU$^+$ macrophages could be observed in all groups, and in agreement with the total cell fluorescence measured in the same region (*Figure 4C*), their number was higher in low- and high-infected individuals compared to PVP-injected controls (*Figure 4D* and corresponding *Video 2*). No significant difference was observed between high- and low-infected fish, confirming that the macrophage number is affected by the presence and not by the number of trypanosomes. Within the trunk region of high-infected larvae, a large proportion of macrophages were observed around and inside the caudal vein, the majority of which were EdU$^+$ (*Figure 4—figure supplement 1A*), suggesting that in high-infected larvae, recently divided macrophages migrated to the vessels. Altogether, these data confirm that *T. carassii* infection triggers macrophage division and that this is higher in infected compared to non-infected individuals, possibly due to a higher haematopoietic activity.

When analysing the neutrophils response, in agreement with the previous observation, the number of neutrophils in the head and trunk regions was not apparently different between infected and non-infected larvae (*Figure 5A*). Neutrophils were scattered throughout the head region, but differently from macrophages, their number did not increase in the area corresponding to the haematopoietic tissue. Quantification of total cell fluorescence in the head and trunk revealed no significant differences between groups (*Figure 5B–C*, *Video 3*). Interestingly, quantification of EdU$^+$ neutrophils in the trunk region, revealed that while in PVP-injected individuals EdU$^+$ neutrophils were rarely observed, in infected fish, a significant, although low number of EdU$^+$ neutrophils was present (*Figure 5D*). These data indicate that neutrophils also respond to the infection by dividing, but their number is relatively low and may not significantly contribute to changes in total cell fluorescence. In contrast to macrophages, within the analysed trunk region, neutrophils were never observed within the posterior cardinal vein or caudal vein, and independently of whether they were EdU$^+$ or not, were mostly observed lining the vessel (*Figure 4—figure supplement 1B*). Altogether, these data indicate that, independent of the trypanosome number, *T. carassii* triggers a differential macrophage

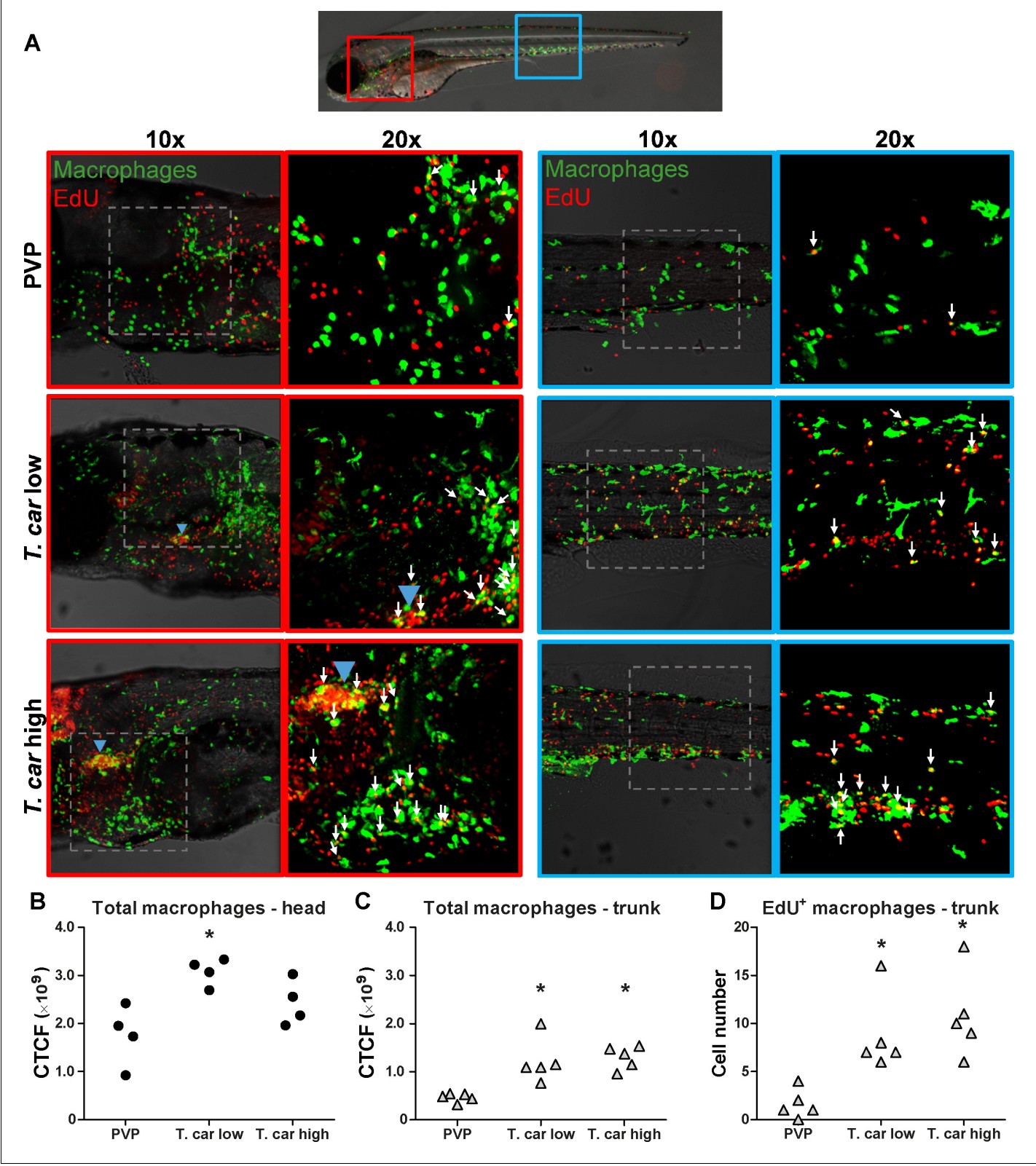

**Figure 4.** *T. carassii* infection triggers macrophage division. (**A**) *Tg(mpeg1:eGFP)* zebrafish larvae were infected intravenously at 5 dpf with n = 200 *T. carassii* or with PVP control. At 3 dpi, larvae received 2 nl 1.13 mM iCLICKTM EdU, at 4 dpi were separated in high- and low-infected individuals and were imaged after fixation and whole mount immunohistochemistry 6–8 hr later (30–32 hr after EdU injection, ~9 dpf). Larvae were fixed and treated with iCLICK EdU ANDY FLUOR 555 (Red) development to identify EdU+ nuclei and with anti-GFP antibody to retrieve the position of macrophages, as
*Figure 4 continued on next page*

*Figure 4 continued*

described in the Materials and methods section. Larvae were imaged with Andor Spinning Disc Confocal Microscope using ×10 and ×20 magnifications. Maximum projections of the head (left panels, red boxes) and trunk (right panels, blue boxes) regions of one representative individual in PVP control, low- and high-infected zebrafish. Images capture macrophages (green) and EdU+ nuclei (red). In the PVP control group, EdU+ nuclei and GFP+ macrophages only rarely overlapped (white arrows, 20x), indicating limited proliferation of macrophages. In high- and low-infected individuals, the number of EdU+ macrophages increased (white arrows, 20x), indicating proliferation of macrophages in response to *T. carassii* infection. Blue arrowhead in the head of low and high-infected larvae, indicates the position of the thymus, an actively proliferating organ at this time point. The identification of EdU+ macrophages (white arrows) was performed upon detailed analysis of the separate stacks used to generate the overlay images, and are provided in *Video 2*. (B–C) Corrected total cell fluorescence (CTCF) calculated in the head (B) and trunk (C) region of larvae described in A. Symbols indicate individual larvae (n = 4–5 per group from two independent experiments). * indicates significant differences to the PVP control as assessed by One-Way ANOVA followed by Bonferroni post-hoc test. (D) *Tg(mpeg1:eGFP)* zebrafish larvae were treated as described in A and the number of EdU+ macrophages in the trunk region of PVP, low- and high-infected larvae was calculated. Symbols indicate individual larvae (n = 5 per group from two independent experiments). * indicates significant differences to the PVP control as assessed by One-Way ANOVA followed by Bonferroni post-hoc test.

The online version of this article includes the following source data and figure supplement(s) for figure 4:

**Source data 1.** *T. carassii* infection triggers macrophage division.

**Figure supplement 1.** Differential distribution of EdU+macrophages and neutrophils along the caudal vein of high- and low-infected larvae.

and neutrophil response, with a significant increase in macrophages number likely due to enhanced myelopoiesis.

## Differential distribution of neutrophils and macrophages in high- and low-infected zebrafish larvae

After having established that *T. carassii* infection triggers macrophage, and to a lesser extent, neutrophil division, we next investigated whether a differential distribution of these cells occurred during infection. Considering that trypanosomes are blood dwelling parasites and the kinetics of parasitaemia, we focused on the caudal vein at 4 dpi, a time point at which clear differences in parasitaemia (*Figure 2*) and a differential distribution of macrophages and neutrophils (*Figures 4–5* and *Figure 4—figure supplement 1*) were observed between high- and low-infected larvae. To this end, crosses between transgenic lines marking the blood vessels and those marking either macrophages or neutrophils were used. *Tg(kdrl:caax-mCherry;mpx:GFP)* or *Tg(fli1:eGFP x mpeg1:mCherry-F)* were infected with *T. carassii*, separated into high- and low-infected larvae at 4 dpi, and imaged with Roper Spinning Disk Confocal Microscope using ×40 magnification. Longitudinal and orthogonal images were analysed to visualise the exact location of cells along the caudal vessels (*Figure 6A* and *Video 4*). In general, macrophages and neutrophils were never observed along or inside the caudal artery allowing us to focus on the caudal vein. In PVP controls, both macrophages and neutrophils were exclusively located outside the caudal vein in close contact with the endothelium or in the tissue adjacent the vessel. In infected fish, while neutrophils remained exclusively outside the vessels (*Figure 6A*, left panel and 7B), macrophages could be seen both inside (white arrows) and outside (blue arrows) the caudal vein (*Figure 6A*, right panel and *Figure 6C*, left plot). Whilst in low-infected individuals macrophage morphology was similar to that observed in non-infected fish, in high-infected larvae, macrophages inside the caudal vein clearly had a more rounded morphology (*Figure 6A*, right panel and *Figure 6D*, right plot). Altogether these data indicate that differently from neutrophils, macrophages increase in number in infected fish, are recruited inside the caudal vein and, depending on the infection level, their morphology can be greatly affected.

## *T. carassii* infection triggers the formation of foamy macrophages in high-infected zebrafish

When analysing macrophage morphology and location, clear differences could be observed between control and high- or low-infected larvae when examined in greater detail. In control fish, macrophages generally exhibited an elongated and dendritic morphology, were very rarely observed inside the caudal vein and were mostly located along the vessel endothelium, in the tissue between the caudal vessels or in the ventral fin (*Figure 7A*, left). A similar morphology and distribution were observed in low-infected larvae (not shown, see also *Figure 6A*). Strikingly, in high-infected larvae, we consistently observed large, dark, granular and round macrophages located

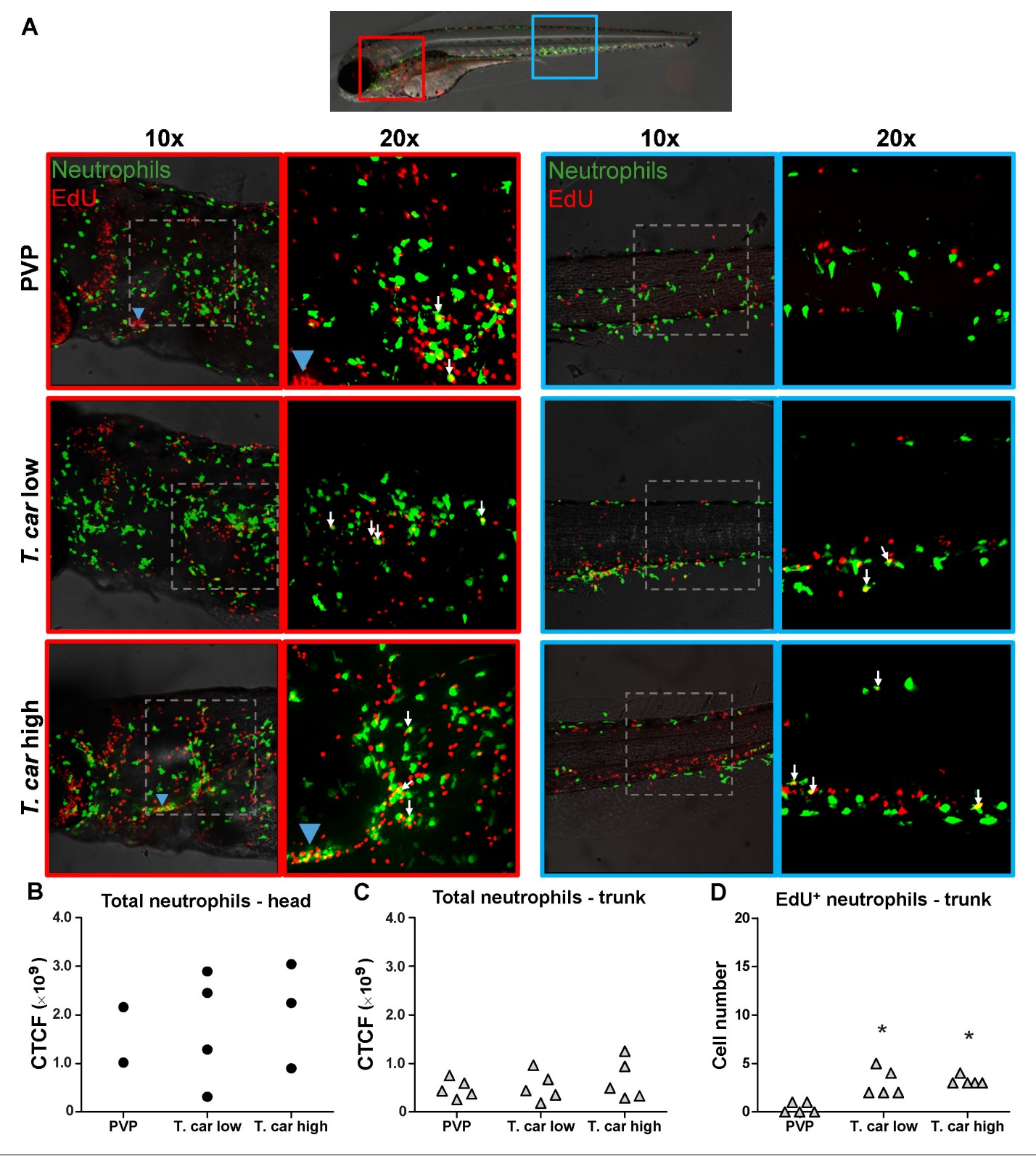

**Figure 5.** *T. carassii* infection triggers neutrophil division. (**A**) *Tg(mpx:GFP)* were treated as described in *Figure 4* (n = 4–5 larvae per group). Maximum projections of the head (left panels, red boxes) and trunk (right panels, blue boxes) regions of one representative individual in PVP, low- and high-infected zebrafish. Images capture neutrophils (green) and EdU+ nuclei (red). The images acquired at a ×20 magnification show that in all groups, EdU+ nuclei and GFP+ neutrophils only rarely overlapped (white arrows), and was marginally higher in infected than in non-infected PVP controls. Detailed

*Figure 5 continued on next page*

*Figure 5 continued*

analysis of the separate stacks selected to compose the overlay image of the head region of the high-infected larva (bottom left panel), revealed that none of the neutrophils in the area indicated by the blue arrowhead (thymus) were EdU$^+$ (*Video 3*). (B–C) Corrected total cell fluorescence (CTCF) calculated in the head (B) and trunk (C) region of larvae described in A. Symbols indicate individual larvae (n = 4–5 per group from two independent experiments). ** indicates significant differences between CTCF in the head and trunk regions, as assessed by Two-Way ANOVA followed by Bonferroni post-hoc test. (D) *Tg(mpx:GFP)* were treated as described in A and the number of EdU$^+$ neutrophils in the trunk region of PVP, low- and high-infected larvae was calculated. Symbols indicate individual larvae (n = 5 per group from two independent experiments). * indicates significant differences to the PVP control as assessed by One-Way ANOVA followed by Bonferroni post-hoc test.

The online version of this article includes the following source data for figure 5:

**Source data 1.** *T. carassii* infection triggers neutrophil division.

inside the caudal vein generally on the dorsal luminal side. These dark macrophages were clearly visible already in bright-field images due to their size, colour, and location, and could be present as single cells or as aggregates (*Figure 7A*, right). The occurrence of these large, granular macrophages increased with the progression of the infection (*Video 5*) and was exclusive to high-infected individuals as they were never observed in low-infected or control larvae.

The rounded morphology, granularity, size and dark appearance of these cells was reminiscent of that of foamy macrophages. Therefore, to further investigate the nature of these cells, the green fluorescent fatty acid BODIPY-FLC5 was used to track lipid accumulation in infected larvae (*Figure 7B*). BODIPY-FLC5 was selected due to its ability to be actively metabolised in de novo triacylglycerides synthesis (*Carten et al., 2011*). This would not only lead to accumulation of the dye in cells with high lipid content, but its accumulation might also be indicative of a change in lipid metabolism, also typical of foamy macrophages. Interestingly, administration of BODIPY-FL5 in infected larvae one day prior to the expected appearance of the large macrophages, revealed the accumulation of lipids in these cells (*Video 6*). Quantification of the number of BODIPY$^+$ and BODIPY$^-$ macrophages, confirmed that BODIPY$^+$ macrophages occur only in high-infected individuals (*Figure 7C*). Macrophages without the large, dark, granular appearance did not show lipid accumulation, independently of the infection level (*Figure 7B*). These results therefore confirms that the large, rounded, granular macrophages in the caudal vein are indeed foamy macrophages.

## Foamy macrophages have a pro-inflammatory activation state

To further investigate the activation state of foamy macrophages, we made use of the *Tg (tnfa:eGFP-F;mpeg1:mCherry-F)* and *Tg(il1b: eGFP-F x mpeg1:mCherry-F)* double transgenic zebrafish lines, having macrophages in red and *tnfa-* or *Il1b*-expressing cells in green (*Figure 8* and *Figure 9*). We first focused on the time point at which the foamy macrophages were most clearly present in highly infected individuals, 4 dpi. Our results clearly show that all large

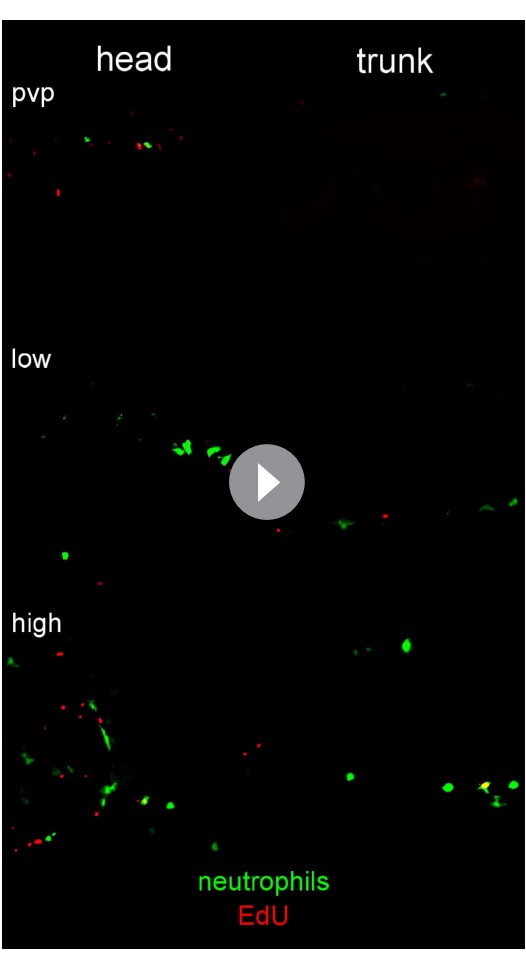

**Video 3.** *T. carassii* infection triggers neutrophils division. AVI files corresponding to the maximum projection images shown in *Figure 6*; Arrows are positioned as in *Figure 6*, and indicate the location of EdU$^+$ neutrophils.
https://elifesciences.org/articles/64520#video3

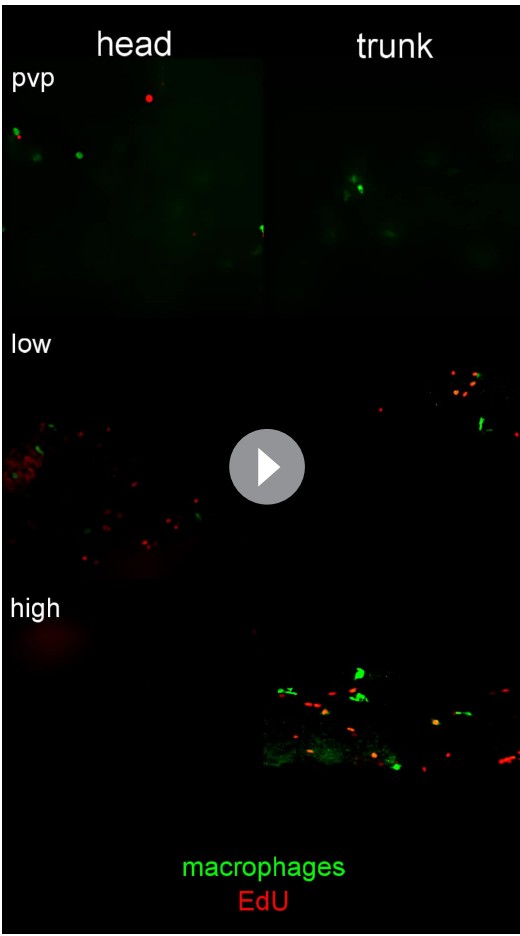

**Video 2.** *T. carassii* infection triggers macrophage division. AVI files corresponding to the maximum projection images shown in *Figure 5*; Arrows are positioned as in *Figure 5*, and indicate the location of EdU⁺ macrophages.
https://elifesciences.org/articles/64520#video2

foamy macrophages, were strongly positive for *tnfa*, suggesting an inflammatory activation state (*Figure 8A*). Interestingly, not only the large foamy macrophages within the caudal vein, but also dendritic or lobulated macrophages outside or lining the vessel showed various degrees of activation. Macrophages that were still partly in the caudal vein (*Figure 8B*, yellow arrowhead) displayed higher *tnfa* expression than macrophages lining the outer endothelium (white arrow heads). This could suggest that the presence of *T. carassii* components within the vessels might trigger macrophage activation.

Similar to what observed for *tnfa* expression, all foamy macrophages within the caudal vein were also positive for *il1b* (*Figure 8C*, asterisk), confirming their pro-inflammatory profile. Interestingly, not only macrophages but also endothelial cells (a selection is indicated by white arrows) were strongly positive for *il1b*. Outside the vessel, cells that were mCherry negative but strongly positive for *il1b* could also be observed (*Figure 8C*, blue arrow); given their position outside the vessel, these are most likely neutrophils. Altogether these data indicate that foamy macrophages occur in high-infected larvae and have a strong pro-inflammatory profile.

## High-infected zebrafish have a strong inflammatory profile associated with susceptibility to infection

When comparing the overall inflammatory state in high- and low-infected larvae it was apparent that high-infected individuals exhibited a higher pro-inflammatory response (*Figure 9*). Although *tnfa*- and *il1b*-positive macrophages could be seen in low-infected individuals, these were generally few (*Figure 9A* and *Figure 9B* middle panels, *Figure 9C*) and a higher number of *tnfa*- and *il1b*-expressing cells was observed in high-infected larvae (*Figure 9A* and *Figure 9B* right panels, *Figure 9C*). In these fish, *il1b* and *tnfa* expression was observed not only in (foamy) macrophages (asterisk), but also in mCherry negative cells outside the vessel (blue arrow, likely neutrophils) and in endothelial cells lining the caudal vein (bright green). To visualise how widespread the inflammatory response was in the embryo, the distribution of *tnfa*-expressing cells was analysed in four different locations spanning the entire trunk and tail of PVP, low- and high-infected individuals (*Figure 9—figure supplement 1*). In these images we appreciate that, in zebrafish as in mammals, *tnfa* expression (eGFP) is not exclusive to immune cells only. In fact, in infected as in non-infected individuals, low constitutive *tnfa* expression is observed in some skin keratinocytes, endothelial cells of the caudal vein and, as previously reported, also enterocytes in the gut villi (*Marjoram et al., 2015*; *Nguyen-Chi et al., 2015*); *tnfa*:eGFP⁺ leukocytes were easily distinguishable by their typical morphology and, in agreements with the distribution of macrophages and neutrophils observed in *Figure 3*, were present mostly in the trunk, distributed along the major vessels, and in the tail, along the tail tip loop and in the fin. Images were acquired with the same settings, thus allowing direct comparison of the intensity of the green signal. The images confirm that *tnfa* expression is strongly inducible in leukocytes and that not only the number of *tnfa*:eGFP⁺ cells but also the intensity of their eGFP signal is higher in high-infected compared to PVP or low-infected individuals. Thus, confirming the overall

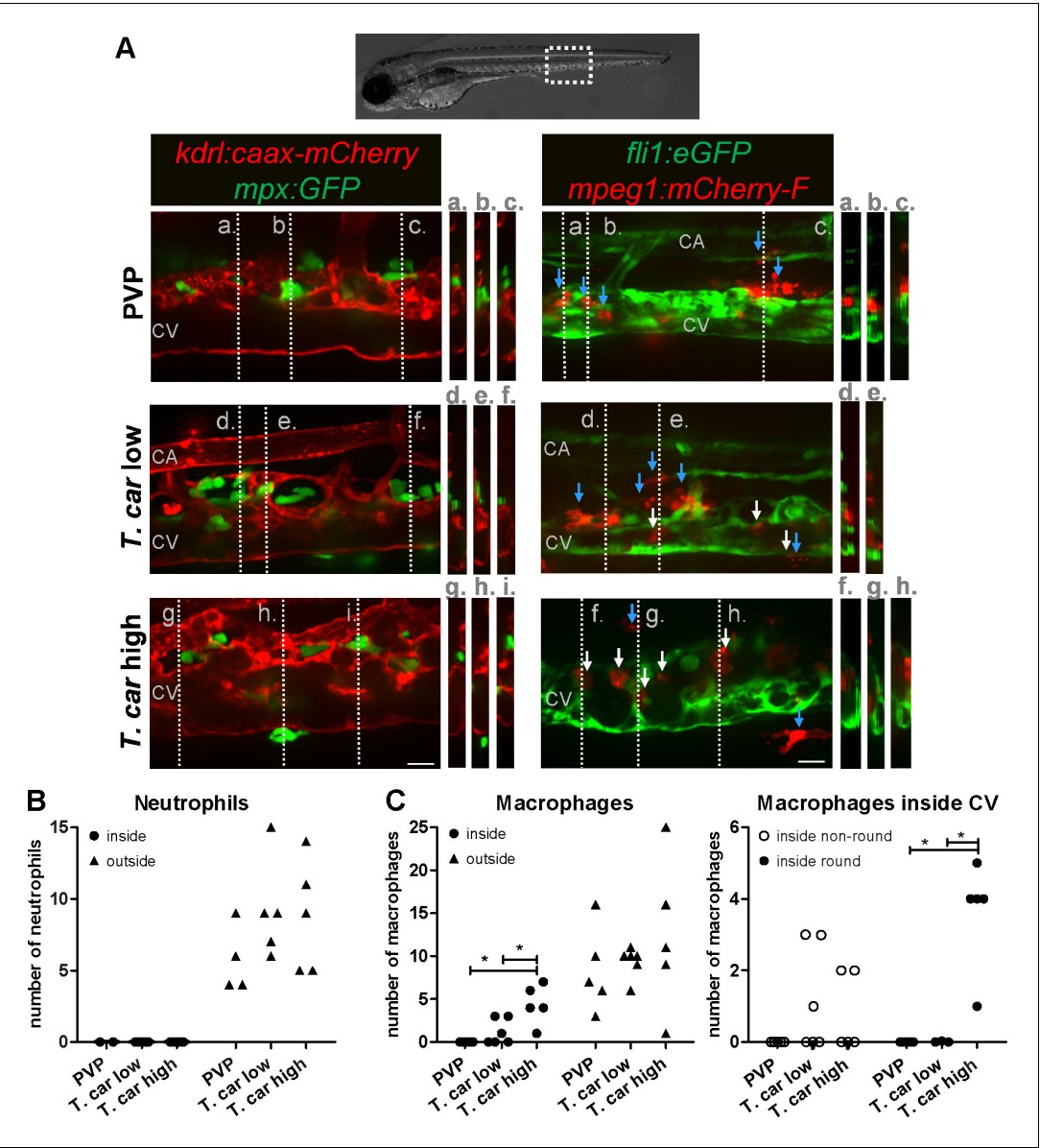

**Figure 6.** Macrophages are recruited into the cardinal caudal vein of high-infected zebrafish larvae. *Tg(kdrl:caax-mCherry;mpx:GFP)* and *Tg(fli1:eGFP x mpeg1:mCherry-F)* zebrafish larvae were injected intravenously at 5 dpf with n = 200 *T. carassii* or with PVP. At 4 dpi larvae were separated in high- and low-infected groups and imaged with a Roper Spinning Disk Confocal Microscope using ×40 magnification. Scale bars indicate 25 μm. CA: caudal artery; CV: caudal vein. (**A**) Left panel: representative images of the longitudinal view of the caudal vessels (red), capturing the location of neutrophils (green). Orthogonal views of the locations marked with grey dashed lines (a,b,c,d,e,f,g,h,i), confirm that in all groups, neutrophils are present exclusively outside the vessels. Right panel: representative images of the longitudinal view of the vessels, capturing the position of macrophages (red) outside the vessels (blue arrowheads) or inside (white arrowheads) the caudal vein (green). Orthogonal views of the locations marked with grey dashed lines (a,b,c,d,e,f,g,h) confirm that in PVP controls, macrophages are present exclusively outside the vessels (blue arrows); in low-infected larvae, most macrophages are outside the vessels (blue arrows) having an elongated or dendritic morphology, although seldomly macrophages can also be observed within the caudal vein (white arrows); in high-infected larvae, although macrophages with dendritic morphology can be seen outside the vessels, the majority of the macrophages resides inside the caudal vein, clearly having a rounded morphology. *Video 4* provides the stacks used for the orthogonal views. (**B–C**) quantification of the number of neutrophils (**B**) and macrophages (**C**) (left panel) inside or outside the caudal vein; of the macrophages observed inside in (**C**), we quantified the number of those with a round or non-round morphology C (right panel). Symbols indicate individual

*Figure 6 continued on next page*

*Figure 6 continued*
larvae (n = 4–6 larvae per group, from two independent experiments). * indicates significant differences as assessed by One-Way ANOVA, followed by Tukey's post-hoc test.
The online version of this article includes the following source data for figure 6:

**Source data 1.** Macrophages are recruited into the cardinal caudal vein of high-infected zebrafish larvae.

higher inflammatory state in high-infected individuals. Similar results were observed using the *il1b: eGFP-F* line (data not shown). Altogether, these results suggest that in high-infected individuals, uncontrolled parasitaemia leads to an exacerbated pro-inflammatory response associated with susceptibility to the infection. Low-infected individuals, however, with moderate *il1b* and *tnfa* responses, are able to control parasitaemia and to recover from the infection.

## Discussion

In this study, we describe the differential response of macrophages and neutrophils in vivo, during the early phase of trypanosome infection of larval zebrafish. Considering the prominent role of innate immune factors in determining the balance between pathology and control of first-peak parasitaemia in mammalian models of trypanosomiasis (*Magez and Caljon, 2011*; *Radwanska et al., 2018*; *Stijlemans et al., 2017*), the use of transparent zebrafish larvae, devoid of a fully developed adaptive immune system, allowed us to investigate the early events of the innate immune response to *T. carassii* infection in vivo. After having established a clinical scoring system of infected larvae, we were able to consistently differentiate high- and low-infected individuals, each associated with opposing susceptibility to the infection. In high-infected larvae, which fail to control first-peak parasitaemia, we observed a strong inflammatory response associated with the occurrence of foamy macrophages and susceptibility to the infection. Conversely, in low-infected individuals, which succeeded in controlling parasitaemia, we observed a moderate inflammatory response associated with resistance to the infection. Altogether these data confirm that also during trypanosome infection of zebrafish, innate immunity is sufficient to control first-peak parasitaemia and that a controlled inflammatory response is beneficial to the host.

Using transgenic lines marking macrophages and neutrophils, total cell fluorescence and cell proliferation analysis revealed that *T. carassii* infection triggers macrophage division, particularly in low-infected individuals. Although to a much lesser extent, neutrophils also responded to the infection by dividing. The total number of neutrophils, however, was comparatively low and likely did not contribute to the total cell fluorescence measured in our whole larvae

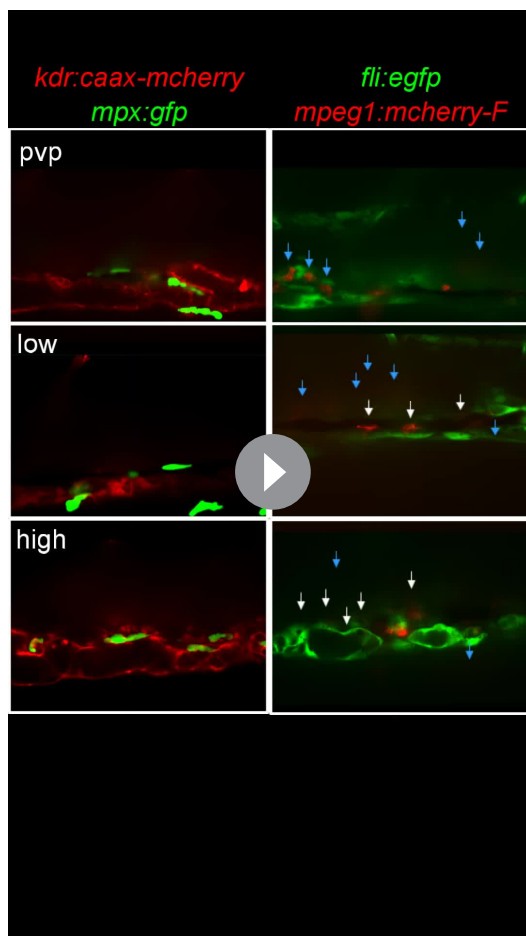

**Video 4.** Macrophages are recruited into the cardinal caudal vein of high-infected zebrafish larvae. Zebrafish larvae were treated and imaged as described in *Figure 7*. Shown are the AVI files corresponding to the maximum projection images shown in *Figure 7*; Neutrophils were never observed within the vessel independently of the infection level (left panels). Macrophages, however, could be seen outside (blue arrows) and inside the vessel (white arrows). The number of rounded macrophages inside the vessel increased with the infection level.
https://elifesciences.org/articles/64520#video4

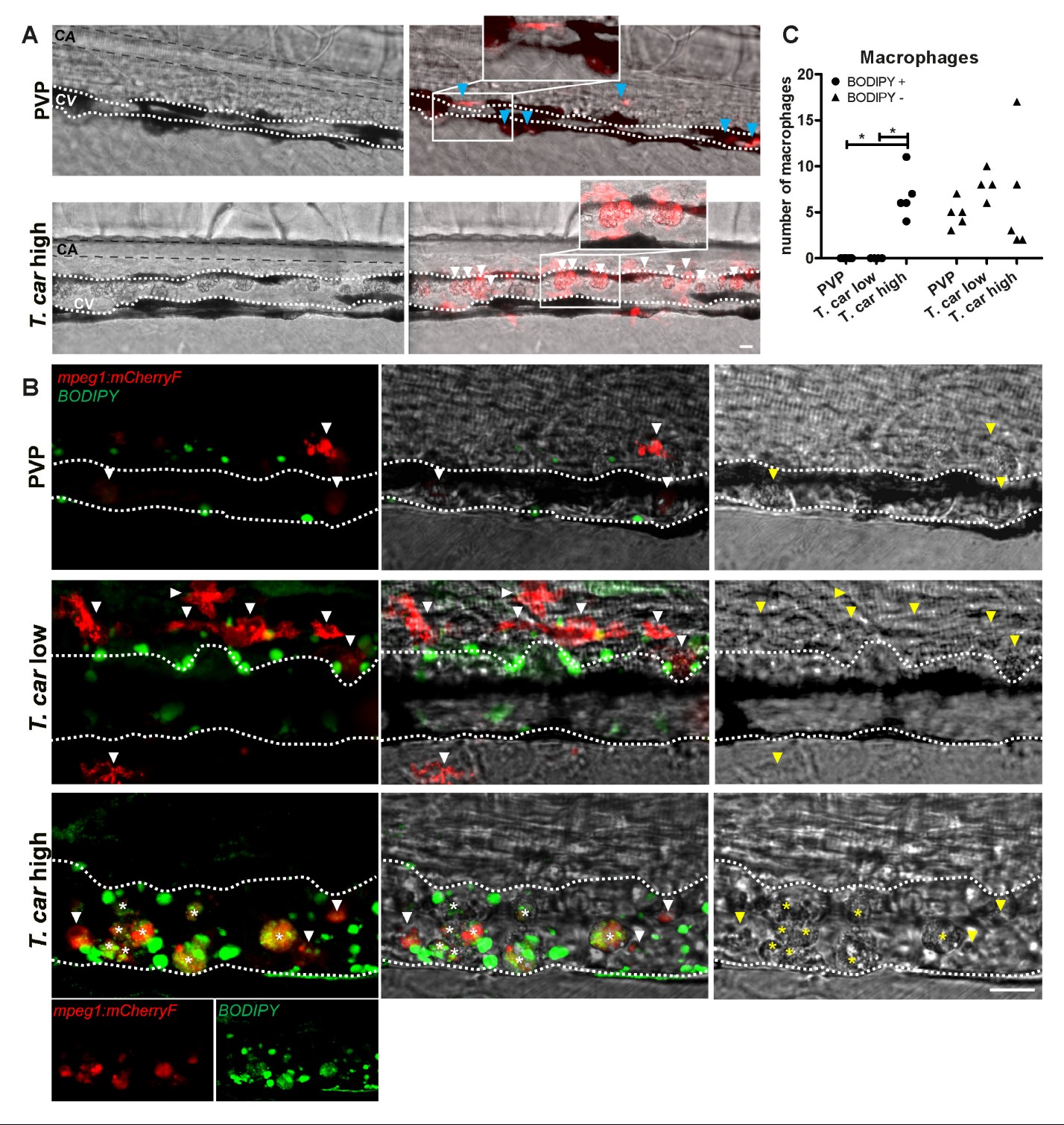

**Figure 7.** The large macrophages inside the caudal vein of high-infected zebrafish are foamy macrophages. (**A**) *Tg(mpeg1:mCherry-F;mpx:GFP)* zebrafish larvae were infected intravenously at 5 dpf with *n* = 200 *T. carassii* or with PVP and imaged at 4 dpi using an Andor Spinning Disc Confocal Microscope at a ×20 magnification. Representative images from three independent experiments are shown, with blue arrowheads pointing at macrophages outside the caudal vein (CV) and white arrowheads indicating large round macrophages inside the caudal vein (white dashed line). Gray dashed line indicated the caudal aorta (CA). Note, how the large macrophages are readily visible in bright field images. Scale bar indicates 25 μm. (**B**) *Tg(mpeg1:mCherry-F)* were treated as in A (n = 5 larvae per group). At 3 dpi, larvae received 1 nl of 30 μM BODIPY-FLC5 and were imaged 18–20 hr later using a Roper Spinning Disc Confocal Microscope at a ×40 magnification. Representative images from three independent experiments are shown.
*Figure 7 continued on next page*

*Figure 7 continued*

*, indicate foamy macrophages: macrophages (red) that are also BODIPY⁺ (green). Note that foamy macrophages are present only in the vein of high-infected individuals. Arrowheads indicate non-foamy macrophages (BODIPY-). Scale bar indicates 25 μm. *Video 6* provides the stacks used in B. (**C**) *Tg (mpeg1:mCherry-F)* were treated as in A and the number of macrophages positive for BODIPY was quantified. BODIPY⁺ macrophages are observed only in high-infected individuals. Symbols indicate individual larvae (n = 4–5 per group, from two independent experiments). * indicates significant differences as assessed by One-Way ANOVA, followed by Tukey's post-hoc test.

The online version of this article includes the following source data for figure 7:

**Source data 1.** The large macrophages inside the caudal vein of high-infected zebrafish are foamy macrophages.

analysis. It cannot be excluded however, that neutrophils' viability was affected by the infection and that the number of newly divided neutrophils is only slightly higher than the dying ones. Although neutrophils were recently implicated in promoting the onset of tsetse fly-mediated trypanosome infections in mouse dermis, macrophage-derived immune mediators, such as NO and TNFα were confirmed to played a more prominent role in the control of first-peak parasitaemia and in the regulation of the overall inflammatory response (*Caljon et al., 2018*).

The observation that in low-infected individuals the number of macrophages was significantly increased by 4–5 dpi, the time point at which clear differences in parasitaemia were apparent between the two infected groups, suggests a role for macrophages, or for macrophage-derived factors in first-peak parasitaemia control. Phagocytosis however, can be excluded as one of the possible contributing factors since motile *T. carassii*, similar to other extracellular trypanosomes (*Caljon et al., 2018*; *Saeij et al., 2003*; *Scharsack et al., 2003*), cannot be engulfed by any innate immune cell (*Video 7*). A strong inflammatory response is also not required for trypanosomes control, since in low-infected individuals, only moderate *il1b* or *tnfα* expression was observed, mostly in macrophages, as assessed using transgenic zebrafish reporter lines. Our data are in agreement with several previous studies using trypanoresistant (BALB/c) or trypanosusceptible (C57Bl/6) mice that revealed the double-edge sword of pro-inflammatory mediators such as TNFα or IFNγ during trypanosome infection in mammalian models (reviewed by *Radwanska et al., 2018*; *Stijlemans et al., 2007*). These studies showed that a timely but controlled expression of IFNγ, TNFα, and NO, contributed to trypanosomes control via direct (*Daulouède et al., 2001*; *Iraqi et al., 2001*; *Lucas et al., 1994*) or indirect mechanisms (*Kaushik et al., 1999*; *Magez et al., 2007*; *Magez et al., 2006*; *Mansfield and Paulnock, 2005*; *Namangala et al., 2001*; *Noël et al., 2002*). Conversely, in individuals in which an uncontrolled inflammatory response

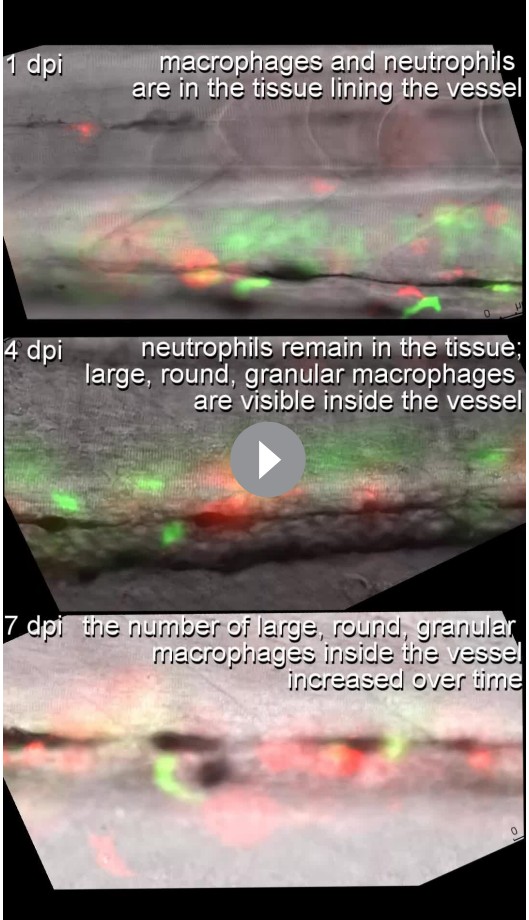

**Video 5.** The occurrence of large granular macrophages increases with the progression of the infection in high-infected individuals. *Tg(mpeg1: mCherry-F;mpx:GFP)* zebrafish larvae were injected intravenously at 5 dpf with n = 200 *T. carassii* or with PVP. At 4 dpi, larvae were separated into high- and low-infected individuals and imaged with a DMi8 inverted digital Leica microscope. The occurrence of large macrophages (arrows) in the cardinal caudal vessel increased with the progression of the infection and was exclusive to high infected individuals (4 and 7 dpi).

https://elifesciences.org/articles/64520#video5

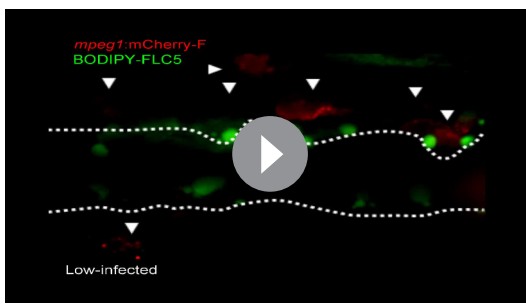

**Video 6.** Large granular macrophages inside the vessel of high-infected larvae are rich in lipid bodies. *Tg (mpeg1:mCherry-F)* zebrafish larvae were infected intravenously at five dpf with n = 200 *T. carassii* or with PVP. At 3 dpi, larvae received 1 nl of 30 µM BODIPY-FLC5 and were imaged 18–20 hr later using a Roper Spinning Disc Confocal Microscope at a ×40 magnification. The AVI files corresponding to the maximum projection images shown in *Figure 8*, as well as a second individual, are shown. Asterisks indicate the position of foamy macrophages inside the caudal vessel (dashed line).

https://elifesciences.org/articles/64520#video6

took place, immunosuppression and inflammation-related pathology occurred (*Namangala et al., 2009*; *Namangala et al., 2001*; *Noël et al., 2004*; *Stijlemans et al., 2016*). The stark contrast between the mild inflammatory response observed in low-infected individuals and the exacerbated response observed in high-infected larvae, strongly resembles the opposing responses generally observed in the aforementioned studies in mice. Owing to the possibility to monitor the infection at the individual level, it was possible to observe such responses within a population of outbred zebrafish larvae. Although we were unable to investigate the specific role of Tnfα during *T. carassii* infection of zebrafish, due to the unavailability of *tnfα-/-* zebrafish lines or the unsuitability of morpholinos for transient knock-down at late stages of development, we previously reported that recombinant zebrafish (as well as carp and trout) Tnfα, are all able to directly lyse *T. brucei* (*Forlenza et al., 2009*). In the same study, we reported that also during *Trypanoplasma borreli* (kinetoplastid) infection of common carp, soluble as well as transmembrane carp Tnfα play a cru-

cial role in both, trypanosome control and susceptibility to the infection. Thus, considering the evolutionary conservation of the lectin-like activity among vertebrate's TNFα (*Daulouède et al., 2001*; *Forlenza et al., 2009*; *Lucas et al., 1994*; *Magez et al., 1997*), it is possible that the direct lytic activity of zebrafish Tnfα may have played a role in the control of first-peak parasitaemia in low-infected individuals. In the future, using *tnfα-/-* zebrafish lines, possibly in combination with *ifnγ* reporter or *ifnγ-/-* lines, it will be possible to investigate in detail the relative contribution of these inflammatory mediators in the control of parasitaemia as well as onset of inflammation.

There are multiple potential explanations for the inability of high-infected larvae to control parasitaemia and the overt inflammatory response. Using various comparative mice infection models, it became apparent that while TNFα production is required for parasitaemia control, a timely shedding of TNFα Receptor-2 (TNFR2) is necessary to limit TNFα-mediated infection-associated immunopathology (*Radwanska et al., 2018*). Furthermore, during *T. brucei* infection in mice and cattle, continuous cleavage of membrane glycosyl-phosphatidyl-inisotol (GPI)-anchored VSG (mVSG-GPI) leads to shedding of the soluble VSG-GIP (sVSG-GPI), while the di-myristoyl-glycerol compound (DMG) is left in the membrane. While the galactose-residues of sVSG-GPI constituted the minimal moiety required for optimal TNFα production, the DMG compound of mVSG contributed to macrophage overactivation (TNFα and IL-1β secretion) (*Magez et al., 2002*; *Magez et al., 1998*; *Sileghem et al., 2001*). Although *T. carassii* was shown to possess a surface dominated by GPI-anchored carbohydrate-rich mucin-like glycoproteins, not subject to antigenic variation (*Lischke et al., 2000*; *Overath et al., 2001*), components of its excreted/secreted proteome, together with phospholipase C-cleaved GPI-anchored surface proteins, have all been shown to play a role in immunogenicity (*Joerink et al., 2007*), inflammation (*Oladiran and Belosevic, 2010*; *Oladiran and Belosevic, 2009*; *Ribeiro et al., 2010*) as well as immunosuppression (*Oladiran and Belosevic, 2012*). Thus, the over-activation caused by the presence of elevated levels of pro-inflammatory trypanosome-derived moieties, combined with the lack of a timely secretion of regulatory molecules (e.g. soluble TNFR2) that could control the host response, may all have contributed to the exacerbated inflammation observed in high-infected individuals.

Given the differential response observed in low- and high-infected individuals, especially with respect to macrophage distribution and activation, we attempted to investigate the specific role of macrophages in the protection or susceptibility to *T. carassii* infection. To this end, the use of a cross between the *Tg(mpeg1:Gal4FF)gl2* (*Ellett et al., 2011*) and the *Tg(UAS-E1b:Eco.NfsB-mCherry)c26*

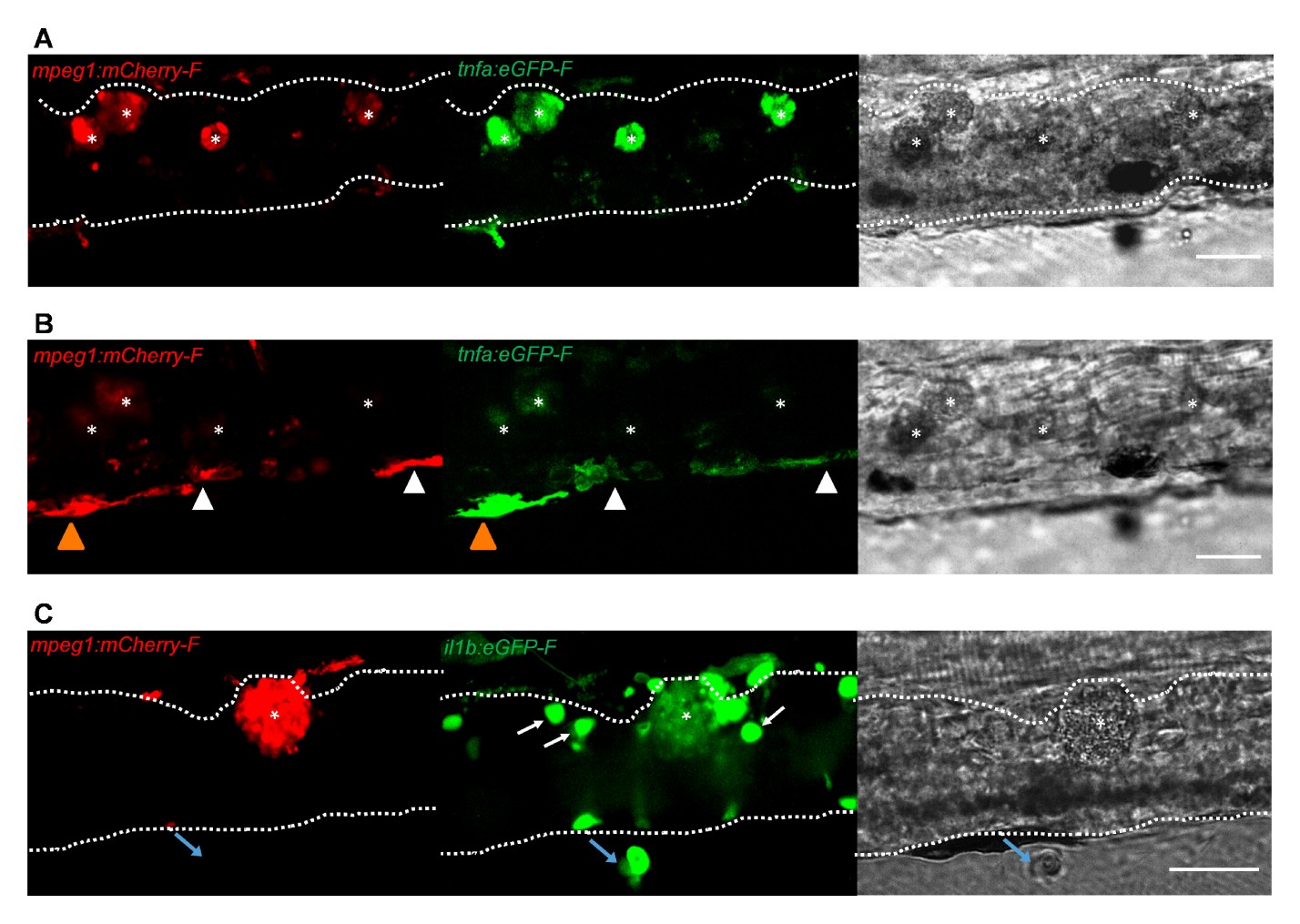

**Figure 8.** Foamy macrophages have an inflammatory profile. *Tg(tnfa:eGFP-F;mpeg1:mCherry-F)* (A-B) or *Tg(il1b:eGFP-F x mpeg1:mcherry-F)* (C) zebrafish larvae (5dpf), were injected with n = 200 *T. carassii* or with PVP. At 4 dpi, high-infected individuals were imaged with an Andor (**A-B**) or Roper (**C**) Spinning Disk Confocal Microscope using ×40 magnification. Scale bar indicates 25 μm. Foamy macrophages (asterisks) were easily identified within the caudal vein (dashed lines) and were strongly positive for *tnfa* (**A**) and *il1b* (**C**) expression (GFP signal). (**B**) Same as A, but a few stacks up, focusing on the cells lining the endothelium. Macrophages that were partly inside and partly outside the vessel (yellow arrowhead) were also strongly positive for *tnfa*, whereas macrophages lining the outer endothelium had a lower *tnfa* expression (white arrowheads). (**C**) A foamy macrophage (asterisk) within the caudal vein (dashed lines) positive for *il1b*. Endothelial cells were also strongly positive for *il1b*, a selection of which is indicated by white arrows. A mCherry-negative-*il1b* positive cell is present outside the vessel (blue arrow). Given its position, it is likely a neutrophil.

(*Davison et al., 2007*) line, which would have allowed the timed metronidazole (MTZ)-mediated depletion of macrophages in zebrafish larvae, was considered. Unfortunately, in vitro analysis of the effect of MTZ on the trypanosome itself, revealed that trypanosomes are susceptible to MTZ, rendering the *nfsB* line not suitable to investigate the role of macrophages (nor neutrophils) during this particular type of infection. Alternatively, we attempted to administer liposome-encapsulated clodronate (Lipo-clodronate) as described previously (*Nguyen-Chi et al., 2017*; *Phan et al., 2018*; *Travnickova et al., 2015*). In our hands, however, administration of 5 mg/ml Lipo-clodronate (3 nl) to five dpf larvae (instead of 2–3 dpf larvae), led to the rapid development of oedema.

Besides differences between the overall macrophage and neutrophil (inflammatory) response, the differential distribution of these cells was also investigated in vivo during infection utilising the transparency of the zebrafish and the availability of transgenic lines marking the vasculature. Neutrophils were never observed inside the cardinal caudal vein although in infected individuals they were certainly recruited and were observed in close contact with the outer vessel's endothelium. Conversely, macrophages could be seen both outside and inside the vessel and the total proportion differed

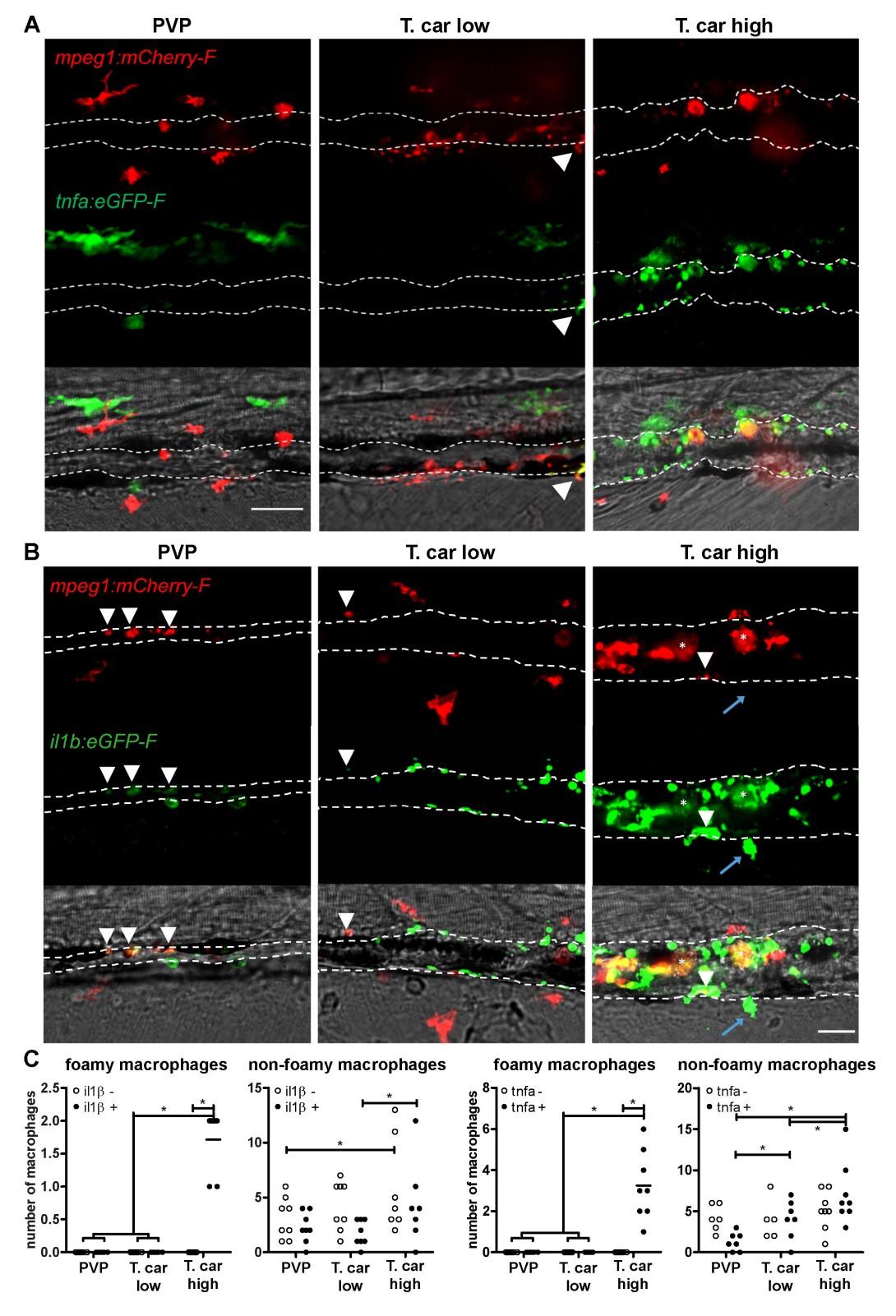

**Figure 9.** High-infected zebrafish have a strong inflammatory profile. Zebrafish larvae (5 dpf), either (**A**) *Tg(tnfa:eGFP-F x mpeg1:mCherry-F)* (n = 8–13 larvae per group from four independent experiments), or (**B**) *Tg(il1b:eGFP-F;mpeg1:mCherry-F)* (n = 7–8 larvae per group from two independent experiments), were infected as described in *Figure 7*. At 4 dpi, larvae were separated in high- and low-infected individuals and imaged with a Roper Spinning Disk Confocal Microscope. Scale bar indicate 25 μm. (**A**) In non-infected PVP controls (left panel), several macrophages can be observed

*Figure 9 continued on next page*

*Figure 9 continued*

outside the vessel but none was positive for *tnfa*. In low-infected individuals (middle panel), macrophages were present inside and outside the vessel. Except the occasional macrophage showing *tnfa*-eGFP expression (white arrowhead), they generally did not exhibit strong eGFP signal. In high-infected individuals however, foamy macrophages (asterisks) as well as endothelial cells (bright green cells) or other leukocytes, were strongly positive for *tnfa*-eGFP expression. (B) *il1b*-eGFP expression was generally low in non-infected PVP controls. In low-infected larvae, *il1b*-positive macrophages were rarely observed (white arrowhead). In both high- and low-infected fish, some endothelium cells in the cardinal caudal vein show high *il1b*-eGFP expression (bright green cells in middle and right panel). In high-infected individual, however (right panel), foamy macrophages inside the vessel (asterisks) as well as other macrophages lining the vessel (white arrowhead) and leukocytes in the tissue (blue arrow), were positive for *il1b*-eGFP expression. (C) Quantification of the total number of foamy and non-foamy macrophages and of the number of those that are positive or not for *il1b* or *tnfa*. All foamy macrophages are positive for *il1b* or *tnfa*, and high-infected individuals have generally a higher number of *il1b* or *tnfa* positive macrophages than low-infected or PVP individuals. *, indicate significant differences as assessed by Two-Way ANOVA followed by Bonferroni post-hoc test.

The online version of this article includes the following source data and figure supplement(s) for figure 9:

**Source data 1.** High-infected zebrafish have a strong inflammatory profile.
**Figure supplement 1.** Differential *tnfa* expression during *T. carassii* infection.

---

between high- and low-infected individuals. While in low-infected individuals the majority of macrophages recruited to the cardinal caudal vein remained outside the vessel in close contact with the endothelium, in high-infected individuals the majority of macrophages were recruited inside the caudal vein and were tightly attached to the luminal vessel wall. To our knowledge, such detailed description of the relative (re)distribution of neutrophils and macrophages, in vivo, during a trypanosome infection, has not been reported before.

Interestingly, exclusively in high-infected individuals, by 4 dpi large, round, dark, and granular cells were observed, already under the bright field view, in the lumen of the cardinal caudal vein. These cells were confirmed to be foamy macrophages with high cytoplasmic lipid content. Foam cells, or foamy macrophages have been named after the lipid bodies accumulated in their cytoplasm leading to their typical enlarged morphological appearance (*Dvorak et al., 1983*), but are also distinguished by the presence of diverse cytoplasmic organelles (*Melo et al., 2003*). Foam cells have been shown to be typical of atherosclerotic plaques associated to various inflammatory metabolic diseases (e.g. hyperlipidemia, diabetes, insulin resistance and obesity) as well as cancer (e.g. Papillary renal cell carcinoma, Esophageal xanthoma and non-small cell lung carcinoma) and autoimmune diseases e.g. multiple sclerosis, systemic lupus erythematosus, rheumatoid arthritis (reviewed in *Guerrini and Gennaro, 2019*; *Saka and Valdivia, 2012*). Besides inflammatory diseases, they have also been associated with several (intracellular) infectious diseases, including Leishmaniasis, Chagas disease, experimental malaria, toxoplasmosis, tuberculosis and other intracellular bacterial infections (reviewed in *Guerrini and Gennaro, 2019*; *López-Muñoz et al., 2018*; *Vallochi et al., 2018*) but never before with (extracellular) trypanosome infection. For example, during *T. cruzi* infection of rat, increased numbers of activated monocytes or macrophages were reported in the blood or heart (*Melo and Machado, 2001*). Interestingly, trypanosome uptake was shown to directly initiate the formation of lipid bodies in macrophages, leading to the appearance of foamy macrophages (*D'Avila et al., 2011*). During human *Mycobacterium tuberculosis* infections, foamy macrophages play a role in sustaining the presence of bacteria and contribute to tissue cavitation enabling the spread of the infection (*Russell et al., 2009*). Independently of the disease, it is clear that foamy macrophages are generally associated with inflammation, since their cytoplasmic lipid bodies are a source of eicosanoids, strong mediators of inflammation (*Melo et al., 2006*;

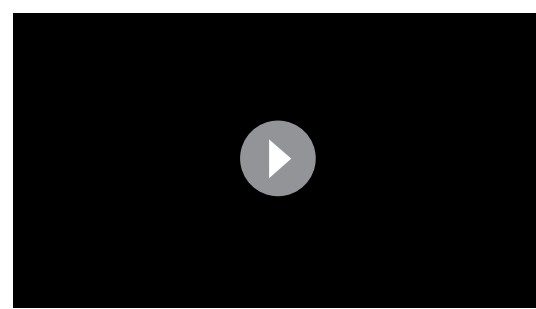

**Video 7.** Motile *T. carassii* cannot be engulfed by innate immune cells. *Tg(mpx:GFP)* zebrafish larvae were injected intravenously at 5 dpf with n = 200 *T. carassii* and two infected individuals were imaged at 7 dpi with a DMi8 inverted digital Leica microscope. Note how motile parasites are relative to static neutrophils (GFP), making it impossible for neutrophils, or any other immune cell, to engulf live parasite.
https://elifesciences.org/articles/64520#video7

*Wymann and Schneiter, 2008*). In turn, inflammatory mediators such as Prostaglandin E2 benefit trypanosome survival, as shown in *Trypanosoma*, *Leishmania*, *Plasmodium*, and *Toxoplasma* infections (reviewed in *Vallochi et al., 2018*). Our results are consistent with these reports as we show the occurrence of foamy macrophages exclusively in individuals that developed high parasitaemia, characterised by a strong pro-inflammatory response, and ultimately succumbed to the infection. Although we did not systematically investigate the exact kinetics of parasitaemia development in correlation with foamy macrophages occurrence, during our in vivo monitoring, we consistently observed that the increase in trypanosome number preceded the appearance of foamy macrophages. It is possible that, in high-infected individuals, foamy macrophages are formed due to the necessity to clear the increasing concentration of circulating trypanosome-derived moieties or of dying trypanosomes. The interaction with trypanosome-derived molecules, including soluble surface (glyco)proteins or trypanosome DNA, may not only be responsible for the activation of pro-inflammatory pathways, but also for a change in cell metabolism. The occurrence of foamy macrophages has been reported for intracellular trypanosomatids (*T. cruzi*, *Leishmania*), and arachidonic acid-derived lipids were reported to act as regulators of the host immune response and trypanosome burden during *T. brucei* infections (*López-Muñoz et al., 2018*). To our knowledge, our study is the first to report the presence of foamy macrophages during an extracellular trypanosome infection.

The possibility to detect the occurrence of large, granular cells already in the bright field and the availability of transgenic lines that allowed us to identify these cells as macrophages, further emphasizes the power of the zebrafish model. It allowed us to visualise in vivo, in real time, not only their occurrence but also their differential distribution with respect to other macrophages or neutrophils. Observations that we might have missed if we for example were to bleed an animal, perform immunohistochemistry or gene expression analysis. Thus, the possibility to separate high- and low-infected animals without the need to sacrifice them, allowed us to follow at the individual level the progression of the infection and the ensuing differential immune response.

In the future, it will be interesting to analyse the transcription profiles of sorted macrophage populations from low- and high-infected larvae. Given the marked heterogeneity in macrophage activation observed especially within high-infected individuals, single-cell transcriptome analysis, of foamy macrophages in particular, may provide insights in the differential activation state of the various macrophage phenotypes. Furthermore, the zebrafish has already emerged as a valuable animal model to study inflammation and host-pathogen interaction and can be a powerful complementary tool to examine macrophage plasticity and polarisation in vivo, by truly reflecting the complex nature of the environment during an ongoing infection in a live host. Finally, the availability of (partly) transparent adult zebrafish lines (*Antinucci and Hindges, 2016*; *White et al., 2008*), may aid the in vivo analysis of macrophage activation in adult individuals.

Altogether, in this study, we describe the innate immune response of zebrafish larvae to *T. carassii* infection. The transparency and availability of various transgenic zebrafish lines, enabled us to establish a clinical scoring system that allowed us to monitor parasitaemia development and describe the differential response of neutrophils and macrophages at the individual level. Interestingly, for the first time in an extracellular trypanosome infection, we report the occurrence of foamy macrophages, characterised by a high-lipid content and strong inflammatory profile, associated with susceptibility to the infection. Our model paves the way to investigate which mediators of the trypanosomes are responsible for the induction of such inflammatory response as well as study the conditions that lead to the formation of foamy macrophages in vivo.

## Materials and methods

**Key resources table**

| Reagent type (species) or resource | Designation | Source or reference | Identifiers | Additional information |
|---|---|---|---|---|
| Gene (*Danio rerio*) | *elongation factor-1α (ef1a)* | DOI:10.7554/eLife.48388 | ZDB-GENE-990415–52 | template for primers for RQ-PCR analysis |
| Gene (*Danio rerio*) | *interleukin-1 beta (il1β)* | ZFIN.org | ZDB-GENE-040702–2 | template for primers for RQ-PCR analysis |

*Continued on next page*

*Continued*

| Reagent type (species) or resource | Designation | Source or reference | Identifiers | Additional information |
|---|---|---|---|---|
| Gene (*Danio rerio*) | *interleukin-10 (il10)* | ZFIN.org | ZDB-GENE-051111–1 | template for primers for RQ-PCR analysis |
| Gene (*Danio rerio*) | *tumor necrosis factor alpha, gene a (tnfa)* | ZFIN.org | ZDB-GENE-050317–1 | template for primers for RQ-PCR analysis |
| Gene (*Danio rerio*) | *tumor necrosis factor alpha, gene b (tnfb)* | ZFIN.org | ZDB-GENE-050601–2 | template for primers for RQ-PCR analysis |
| Gene (*Danio rerio*) | *interleukin-6 (il6)* | ZFIN.org | ZDB-GENE-120509–1 | template for primers for RQ-PCR analysis |
| Gene (*Trypanosoma carassii*) | *heat-shock protein-70 (hsp70)* | DOI:10.7554/eLife.48388 | GeneBank-FJ970030.1 | template for primers for RQ-PCR analysis |
| Strain, strain background (*Cyprinus carpio*) | Wild type common carp, R3xR8 strain | DOI:10.1016/0044-8486 (95)91961 T | | used to passage *Trypanosoma carassii* in vivo |
| Strain, strain background (*Danio rerio*) | Wild type zebrafish, AB strain | Zebrafish International Resource Center | RRID:SCR_005065; Cat#ZL1 | used for backcrossing of all Tg |
| Strain, strain background (*Danio rerio*) | casper strain | DOI:10.1016/j.stem.2007.11.002 | | optically transparent |
| Strain, strain background (*Danio rerio*) | AB:Tg(mpx:GFP)[i114] | DOI:10.1182/blood-2006-05-024075 | | wild type line marking neutrophils with green fluorescent protein (GFP) under the control of the *mpx* (myeloperoxidase) promotor |
| Strain, strain background (*Danio rerio*) | *Tg(mpeg1:mCherry-F)[ump2Tg]* | DOI:10.1242/dmm.014498 | | wild type line marking macrophages with farnesylated red fluorescent protein (mCherry) under the control of the mpeg1 (Macrophage expressed gene-1) promotor |
| Strain, strain background (*Danio rerio*) | Tg(*mpeg1:eGFP*)[gl22] | DOI:10.1182/blood-2010 -10-314120 | | wild type line marking macrophages with green fluorescent protein (GFP) under the control of the mpeg1 (Macrophage expressed gene-1) promotor |
| Strain, strain background (*Danio rerio*) | *AB:Tg(kdrl:caax-mCherry)* | DOI:10.1101/gad.1629408.734 | | wild type line marking the vasculature with green fluorescent protein (GFP) under the control of the *kdrl* (Vascular endothelial growth factor receptor kdr-like) promotor. Old name: Tg(*flk1:ras-cherry*)[s896] |
| Strain, strain background (*Danio rerio*) | *casper Tg(fli:egfp)[y1]* | DOI:10.1038/nrg888 | | optically transparent line, marking the vasculature with green fluorescent protein (GFP) under the control of the endothelial cell marker fli1 (friend leukemia integration-1) promotor |
| Strain, strain background (*Danio rerio*) | *Tg(il1b:eGFP-F)[ump3Tg]* | DOI:10.1242/dmm.014498 | | wild type line marking tnfa-expressing cells with farnesylated green fluorescent protein (GFP-F) under the control of the zebrafish tnfa (tumor necrosis factor alpha a) promotor |

*Continued on next page*

*Continued*

| Reagent type (species) or resource | Designation | Source or reference | Identifiers | Additional information |
|---|---|---|---|---|
| Strain, strain background (*Danio rerio*) | *Tg(tnfa:eGFP-F)*[ump5Tg] | DOI:10.7554/eLife.07288 | | wild type line marking il1b-expressing cells with farnesylated green fluorescent protein (GFP-F) under the control of the zebrafish il1b (interleukin 1-beta) promotor |
| Strain, strain background (*Trypanosoma carassii*) | TsCc-NEM strain | doi:10.1007/s004360050408 | | |
| Antibody | Chicken polyclonal anti-GFP | Aves Labs | Cat# GFP-1010, RRID:AB_2307313 | primary antibody, whole mount: 1:500 |
| Antibody | Goat polyclonal anti-chicken-Alexa 488 | Abcam | Cat# ab150169, RRID:AB_2636803 | Secondary antibody, whole mount: 1:500 |
| Chemical compound, drug | BODIPYTM FL pentanoic acid | Invitrogen | BODIPY-FL5: Cat# D-3834 | |
| Commercial assay or kit | iCLICKTM EdU (5- ethynyl-2'- deoxyuridine, component A) | ABP Biosciences | ANDY FLUOR 555 Imaging Kit: Cat# A004 | |

## Zebrafish lines and maintenance

Zebrafish were kept and handled according to the Zebrafish Book (zfin.org) and local animal welfare regulations of The Netherlands. Zebrafish embryo (0–5 days post fertilisation (dpf)) were raised at 27°C with a 12:12 light-dark cycle in egg water (0.6 g/L sea salt, Sera Marin, Heinsberg, Germany) and at five dpf transferred to E2 water (NaCl 15 mM, KCl 0.5 mM, MgSO4 1 mM, $KH_2PO_4$0.15 mM, $Na_2HPO_4$0.05 mM, CaCl 1 mM, $NaHCO_3$0.7 mM). From 5 days post fertilisation (dpf) until 14 dpf larvae were fed Tetrahymena once a day. From 10 dpf, larvae were also daily fed dry food ZM-100 (ZM systems, UK). The following zebrafish lines or crosses thereof were used in this study: transgenic *Tg(mpx:GFP)*[i114] (*Renshaw et al., 2006*) marking neutrophils, *Tg(kdrl:hras-mCherry)*[s896] referred as *Tg(kdrl:caax-mCherry)* (*Chi et al., 2008*; *Jin et al., 2005*) and *Tg(fli1:eGFP)*[y1] (*Lawson and Weinstein, 2002*) marking the vasculature, *Tg(mpeg1:eGFP)*[gl22] (*Ellett et al., 2011*) and *Tg(mpeg1:mCherry-F)*[ump2Tg] marking macrophages, *Tg(il1b:eGFP-F)*[ump3Tg], (*Nguyen-Chi et al., 2014*), *Tg(tnfa:eGFP-F)*[ump5Tg](*Nguyen-Chi et al., 2015*) marking cytokine-expressing cells. The latter three transgenic zebrafish lines express a farnesylated (membrane-bound) mCherry (mCherry-F) or eGFP (eGFP-F) under the control of the *mpeg1, il1b* or *tnfa* promoter, respectively. *All lines have a AB (wild type) background except for the Tg(fli1:eGFP)*[y1] which was kept as optically transparent casper line (*White et al., 2008*) and crossed with the specified lines.

## *Trypanosoma carassii* culture and infection of zebrafish larvae

*Trypanosoma carassii* (strain TsCc-NEM) was cloned and characterised previously (*Overath et al., 1998*) and maintained in our laboratory by syringe passage through common carp (*Cyprinus carpio*) as described previously (*Dóró et al., 2019*). Blood was drawn from infected carp and kept at 4°C overnight in siliconised tubes. Trypanosomes enriched at the interface between the red blood cells and plasma were collected and centrifuged at 800 xg for 8 min at room temperature. Trypanosomes were resuspended at a density of $5 \times 10^5$–$1\times10^6$ ml and cultured in 75 or 165 cm$^2$ flasks at 27°C without $CO_2$ in complete medium as described previously (*Dóró et al., 2019*). *T. carassii* were kept at a density below $5 \times 10^6$/ml, and sub-cultured 1–3 times a week. In this way *T. carassii* could be kept in culture without losing infectivity for up to 2 months. The majority of carp white blood cells present in the enriched trypanosome fraction immediately after isolation, died within the first 3–5 days of culture and any remaining blood cell was removed prior to *T. carassii* injection into zebrafish. To this end, cells were centrifuged at 800 xg for 5 min in a 50 ml Falcon tube and the tube was subsequently tilted in a 20° angle in relation to the table surface, facilitating the separation of the motile trypanosomes along the conical part of the tube from the static pellet of white blood cells at the bottom of the tube.

For zebrafish infection, trypanosomes were cultured for 1 week and no longer than 3 weeks. Infection of zebrafish larvae was performed as described previously (*Dóró et al., 2019*). Briefly, prior to injection, 5 days post fertilisation (dpf) zebrafish larvae were anaesthetised with 0.017% Ethyl 3-aminobenzoate methanesulfonate (MS-222, Tricaine, Sigma-Aldrich) in egg water. *T. carassii* were resuspended in 2% polyvinylpyrrolidone (PVP, Sigma-Aldrich) and injected (n = 200) intravenously in the Duct of Cuvier. After injection, infected and non-infected larvae were kept in separate tanks at a density of 60 larvae per 1L water.

### Clinical scoring system of the severity of infection

Careful monitoring of the swimming behaviour of zebrafish larvae after infection (five dpf onwards) as well as in vivo observation of parasitaemia development in transparent larvae, led to the observation that from 4 dpi onwards larvae could generally be segregated into high- and low-infected individuals. To objectively assign zebrafish to either one of these two groups, we developed a clinical scoring system (*Figure 10*). The first criterion looked at the escape reflex upon contact with a pipette and was sufficient to identify high-infected individuals as those not reacting to the pipet (slow swimmers). To categorise the remaining individuals, a second criterion based on counting parasite:blood cell ratios in 100 events passing through the intersegmental capillary (ISC) above the cloaca was developed. The infection scores on a scale from 1 to 10 were assigned as follows: 1 = no parasites observed, 2 = 1–10% parasite, 3 = 11–20% parasite, 4 = 21–30% parasite, 5 = 31–40% parasite, 6 = 41–55% parasite, 7 = 56–70% parasite, 8 = 71–85% parasite, 9 = 86–99%, 10 = no blood cells observed. Larvae with infection scores between 1–3 were categorised as low-infected while scores between 6 and 10 were categorised as high infected. Larvae with scores 4–5 were reassessed 1 day later, at 5 dpi, and then categorised as high- or low-infected. Since handing may affect larval behaviour or overall gene expression, larvae with scores 4–5 were re-assessed at 5 dpi only when image analysis was performed and not when survival or gene expression analysis were carried out. Larvae with a score of 1 (no parasites observed in ISC) where immediately assessed using the third criterion (extravasation, see below) and were never found to be parasite-free. Thus, remained

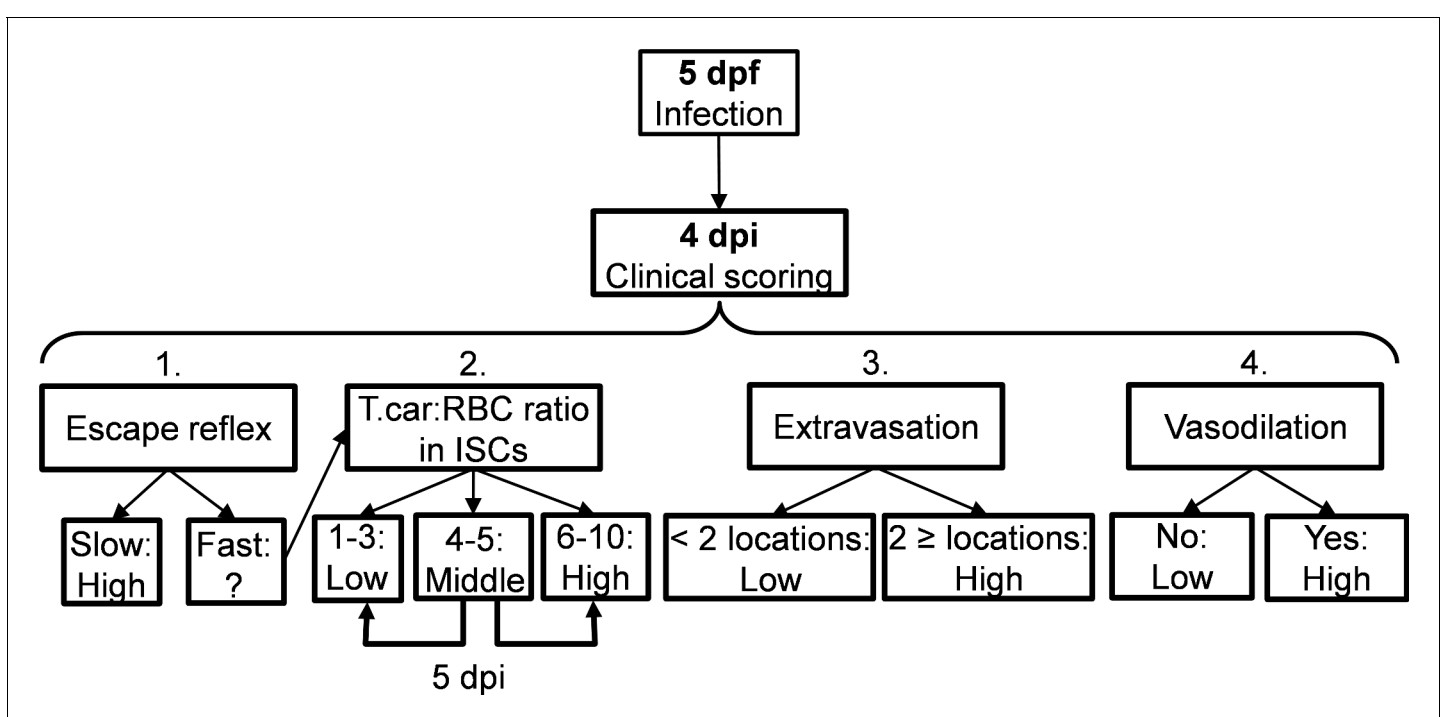

**Figure 10.** Schematic overview of the clinical scoring system used to determine individual infection levels of *T.carassii*-infected zebrafish larvae. Zebrafish larvae infected with *T. carassii* can be analysed at 4 dpi; based on up to four different parameters including (1) visual monitoring of larval behaviour, (2) parasite numbers, (3) location, or (4) vasodilation, larvae could be segregated into high- and low-infected individuals. See details in the text in the corresponding Materials and methods section.

assigned to the low-infected group. Next to that, heartbeat of the larvae was monitored and noted if it was slower than the control. The third criterion considered extravasation and the location of extravasated parasites (e.g. fins, muscle, intraperitoneal cavity, and interstitial space lining the blood vessels). Finally, for the fourth criterion the diameter of the caudal vein in the trunk area after the cloaca region was measured in ImageJ 1.49o to quantify the degree of vasodilation. Eventual blockage of tail tip vessel-loop was also noted. In general, the swimming behaviour of larvae was observed and compared to the control group.

### Real-time quantitative PCR

Zebrafish were sacrificed by an overdose of MS-222 anaesthetic (50 mg/L). At each time point, 3–6 zebrafish larvae were sacrificed and pooled. The control at time point zero, was a group of larvae injected with $n = 200$ *T. carassii* and immediately sacrificed. Pools were transferred to RNA later (Ambion), kept at 4℃ overnight and then transferred to −20℃ for further storage. Total RNA isolation was performed with the Qiagen RNeasy Micro Kit (QIAgen, Venlo, The Netherlands) according to manufacturer's protocol. Next, 250–500 ng total RNA was used as template for cDNA synthesis using SuperScript III Reverse Transcriptase and random hexamers (Invitrogen, Carlsbad, CA, USA), following the manufacturer's instructions with an additional DNase step using DNase I Amplification Grade (Invitrogen, Carlsbad, CA, USA). cDNA was then diluted 25 times to serve as template for real-time quantitative PCR (RT-qPCR) using Rotor-Gene 6000 (Corbett Research, QIAgen), as previously described (*Forlenza et al., 2012*). Primers (*Table 1*) were obtained from Eurogentec (Liège, Belgium). Gene expression was normalised to the expression of *elongation factor-1 alpha* (*ef1a*) housekeeping gene and expressed relative to the *T. carassii*-injected control at 0 dpi.

### In vivo imaging and videography of zebrafish

Prior to imaging, zebrafish larvae were anaesthetised with 0.017% MS-222 (Sigma-Aldrich). For total fluorescence acquisition, double transgenic *Tg(mpeg1:mCherry-F;mpx:GFP)* were positioned on preheated flat agar plates (1% agar in egg water with 0.017% MS-222) and imaged with Fluorescence Stereo Microscope (Leica M205 FA). The image acquisition settings were as following: Zoom: 2.0–2.2, Gain: 1, Exposure time (ms): 70 (BF)/700 (GFP)/1500 (mCherry), Intensity: 60 (BF)/700 (GFP)/700 (mCherry), Contrast: 255/255 (BF)/70/255 (GFP)/15/255 (mCherry).

Alternatively, anaesthetised larvae were embedded in UltraPure LMP Agarose (Invitrogen) and positioned on the coverglass of a 35 mm petri dish, (14 mm microwell, coverglass No. 0 (0.085–0.13 mm), MatTek corporation) prior to imaging. A Roper Spinning Disk Confocal (Yokogawa) on Nikon Ti Eclipse microscope with 13 × 13 Photometrics Evolve camera (512 × 512 Pixels 16 × 16 micron) equipped with a 40x (1.30 NA, 0.24 mm WD) OI objective, was used with the following settings: GFP excitation: 491 nm, emission: 496–560 nm, digitizer: 200 MHz (12-bit); 561 BP excitation: 561 nm; emission: 570–620 nm, digitizer: 200 MHz (12-bit); BF: digitizer: 200 MHz (12-bit). Z-stacks of 1 or 0.5 µm. An Andor-Revolution Spinning Disk Confocal (Yokogawa) on a Nikon Ti Eclipse microscope with Andor iXon888 camera (1024 × 1024 Pixels 13 × 13 micron) equipped with 40x (0.75 NA, 0.66 mm WD) objective, 40x (1.15 NA, 0.61–0.59 mm WD) WI objective, 20x (0.75 NA, 1.0 mm WD) objective and 10x (0.50 NA, 16 mm WD) objective was used with the following settings: dual pass 523/561: GFP excitation: 488 nm, emission: 510–540 nm, EM gain: 20–300 ms, digitizer: 10 MHz (14-bit); RFP excitation: 561 nm; emission: 589–628 nm, EM gain: 20–300 ms, digitizer: 10 MHz

**Table 1.** List of primers used in this study.

| Gene name | Fw primer sequence | RV primer sequence | Acc. number (zfin.org) |
|---|---|---|---|
| *ef1a* | CTGGAGGCCAGCTCAAACAT | ATCAAGAAGAGTAGTAGTACCG | ZDB-GENE-990415–52 |
| *il1β* | TTGTGGGAGACAGACAGTGC | GATTGGGGTTTGATGTGCTT | ZDB-GENE-040702–2 |
| *il10* | ACTTGGAGACCATTCTGCC | CACCATATCCCGCTTGAGTT | ZDB-GENE-051111–1 |
| *tnfa* | AAGTGCTTATGAGCCATGC | CTGTGCCCAGTCTGTCTC | ZDB-GENE-050317–1 |
| *tnfb* | AAACAACAAATCACCACACC | ACACAAAGTAAAGACCATCC | ZDB-GENE-050601–2 |
| *il6* | ACTCCTCTCCTCAAACCT | CATCTCTCCGTCTCTCAC | ZDB-GENE-120509–1 |
| *T. car. hsp70* | CAGCCGGTGGAGCGCGT | AGTTCCTTGCCGCCGAAGA | FJ970030.1 (GeneBank) |

(14-bit); BF DIC EM gain: 20–300 ms, digitizer: 10 MHz (14-bit). Z-stacks of 1 µm. A Zeiss lsm-510 confocal microscope equipped with 20x long-distance objective was used with the following settings: laser excitation = 488 nm with 73% transmission; HFT filter = 488 nm; BP filter = 505–550; detection gain = 800; amplifier offset = −0.01; amplifier gain = 1.1; bright field channel was opened with Detection Gain = 130; frame size (pixels) = 2048×2048; pinhole = 300 (optical slice < 28.3 µm, pinhole ø = 6.26 airy units). Images were analysed with ImageJ-Fijii (version 1.52 p).

High-speed videography of *T. carassii* swimming behaviour in vivo was performed as described previously (*Dóró et al., 2019*). Briefly, the high-speed camera was mounted on a DMi8 inverted digital microscope (Leica Microsystems), controlled by Leica LASX software (version 3.4.2.) and equipped with 40x (NA 0.6) and 20x (NA 0.4) long distance objectives (Leica Microsystems). For high-speed light microscopy a (eight bits) EoSens MC1362 (Mikrotron GmbH, resolution 1280 × 1024 pixels), with Leica HC 1x Microscope C-mount Camera Adapter, was used, controlled by XCAP-Std software (version 3.8, EPIX inc). Images were acquired at a resolution of 900 × 900 or 640 × 640 pixels. Zebrafish larvae were anaesthetised with 0.017% MS-222 and embedded in UltraPure LMP Agarose (Invitrogen) on a microscope slide (1.4–1.6 mm) with a well depth of 0.5–0.8 mm (Electron Microscopy Sciences). Upon solidification of the agarose, the specimen was covered with 3–4 drops of egg water containing 0.017% MS-222, by a 24 × 50 mm coverslip and imaged immediately. For all high-speed videography, image series were acquired at 480–500 frames per second (fps) and analysed using a PFV software (version 3.2.8.2) or MiDAS Player v5.0.0.3 (Xcite, USA).

## Fluorescence quantification

Quantification of total cell fluorescence in zebrafish larvae was performed in ImageJ (version 1.49o) using the free-form selection tool and by accurately selecting the larvae area. Owing to the high auto-fluorescence of the gut or gut content, and large individual variation, the gut area was excluded from the total fluorescence signal. Area integrated intensity and mean grey values of each selected larva were measured by the software. To correct for the background, three consistent black areas were selected in each image. Analysis was performed using the following formula: corrected total cell fluorescence (CTCF) = Integrated density – (Area X Mean background value).

## EdU proliferation assay and immunohistochemistry

iCLICK EdU (5- ethynyl-2'- deoxyuridine, component A) from ANDY FLUOR 555 Imaging Kit (ABP Biosciences) at a stock concentration of 10 mM, was diluted in PVP to 1.13 mM. Infected *Tg(mpeg1: eGFP)* or *Tg(mpx:GFP)* larvae were injected in the heart cavity at 3 dpi (8dpf) with 2 nl of solution, separated in high- and low-infected individuals at 4 dpi and euthanised 6–8 hr later (30–32 hr after EdU injection) with an overdose of anaesthetic (0.4% MS-222 in egg water). Following fixation in 4% paraformaldehyde (PFA, Thermo Scientific) in PBS, o/n at 4°C, larvae were washed three times in buffer A (0.1% (v/v) tween-20, 0.05% (w/v) azide in PBS), followed by dehydration: 50% MeOH in PBS, 80% MeOH in H$_2$0 and 100% MeOH, for 15 min each, at room temperature (RT), with gentle agitation. To remove background pigmentation, larvae were incubated in bleach solution (5% (v/v) H$_2$O$_2$ and 20% (v/v) DMSO in MeOH) for 1 hr at 4°C, followed by rehydration: 100% MeOH, 80% MeOH in H2O, 50% MeOH in PBS for 15 min each, at room temperature (RT), with gentle agitation. Next, larvae were incubated three times for 5 min each in buffer B (0.2%(v/v) triton-x100, 0.05% azide in PBS) at RT with gentle agitation followed by incubation in EdU iCLICK development solution for 30 min at RT in the dark and three rapid washes with buffer B.

The described EdU development led to loss of GFP signal in the transgenic zebrafish. Therefore, to retrieve the position of neutrophils or macrophages, wholemount immunohistochemistry was performed. Larvae were blocked in 0.2% triton-x100, 10% DMSO, 6% (v/v) normal goat serum and 0.05% azide in PBS, for 3 hr, at RT with gentle agitation. Next, the primary antibody Chicken anti-GFP (Aves labs.Inc, 1:500) in Antibody buffer (0.2% tween-20, 0.1% heparin, 10% DMSO, 3% normal goat serum and 0.05% azide in PBS) was added and incubated overnight (o/n) at 37°C. After three rapid and three 5 min washes in buffer C (0.1% tween-20, 0.1% (v/v) heparin in PBS), at RT with gentle agitation, the secondary antibody goat anti-chicken-Alexa 488 (Abcam, 1:500) was added in Antibody buffer and incubated o/n at 37°C. After three rapid and three 5 min washes in buffer C, at RT with gentle agitation, larvae were imaged with Andor Spinning Disk Confocal Microscope.

## BODIPY injection

BODIPY FL pentanoic acid (BODIPY-FL5, Invitrogen) was diluted in DMSO to a 3 mM stock solution. Stock solution was diluted 100x (30 µM) with PVP. Infected larvae 3 dpi (8 dpf) were injected with 1 nl of the solution i.p. (heart cavity) and imaged 18–20 hr later.

## Statistical analysis

Analysis of gene expression and total fluorescence data were performed in GraphPad PRISM 5. Statistical analysis of gene expression data was performed on Log(2) transformed values by One-way ANOVA followed by Tukey's or Bonferroni multiple comparisons test. Analysis of Corrected Total Cell Fluorescence was performed on Log(10) transformed values followed by Two-Way ANOVA and Bonferroni multiple comparisons post-hoc test. Analysis of EdU$^+$ macrophages was performed on Log(10) transformed values followed by One-way ANOVA and Bonferroni multiple comparisons post-hoc test. In all cases, $p<0.05$ was considered significant.

## Acknowledgements

Dr. Christelle Langevin from Institut National de la Recherche Agronomique (INRA) is greatly acknowledged for her assistance with immunohistochemical analysis (IHC). IHC analyses benefited from the expertise of the Fish phenotyping platform IERP-UE907, Jouy-en-Josas Research Center, France DOI: 10.15454/1.5572427140471238E12 belonging to the National Distributed Research Infrastructure for the Control of Animal and Zoonotic Emerging Infectious Diseases through In Vivo Investigation (EMERG'IN DOI: 10.15454/1.5572352821559333E12). The authors like to thank Dr. Danilo Pietretti, Marleen Scheer, and Dr. Sylvia Brugman from the Cell Biology and Immunology Group of Wageningen University and Research (WUR) for technical support with the RQ-PCR analysis, parasite isolation and for fruitful discussions; the CARUS Aquatic Research Facility of WUR is acknowledged for fish rearing and husbandry. Prof. Mark Carrington, Cambridge University, is acknowledged for the fruitful discussions and for revising the manuscript. Furthermore, the authors like to thank Dr. Norbert de Ruijter from the Wageningen Light Microscopy Centre, and the Montpellier Resources Imagerie facility for their assistance.

## Additional information

### Funding

| Funder | Grant reference number | Author |
| --- | --- | --- |
| European Union 7th Framework Programme | PITN-GA-2011-289209 | Eva Dóró |
| Nederlandse Organisatie voor Wetenschappelijk Onderzoek | 022.004.005 | Sem H Jacobs |

The funders had no role in study design, data collection and interpretation, or the decision to submit the work for publication.

### Author contributions

Sem H Jacobs, Conceptualization, Data curation, Formal analysis, Visualization, Methodology, Writing - original draft, Writing - review and editing; Eva Dóró, Data curation, Formal analysis, Visualization, Methodology, Writing - review and editing; Ffion R Hammond, Formal analysis, Methodology, Writing - review and editing; Mai E Nguyen-Chi, Georges Lutfalla, Conceptualization, Supervision, Methodology, Writing - review and editing; Geert F Wiegertjes, Conceptualization, Supervision, Funding acquisition, Writing - original draft, Project administration, Writing - review and editing; Maria Forlenza, Conceptualization, Data curation, Formal analysis, Supervision, Funding acquisition, Visualization, Methodology, Writing - original draft, Project administration, Writing - review and editing

## Author ORCIDs
Sem H Jacobs https://orcid.org/0000-0001-7482-3438
Maria Forlenza https://orcid.org/0000-0001-9026-7320

## Ethics
Animal experimentation: All animals were handled in accordance with good animal practice as defined by the European Union guidelines for handling of laboratory animals (http://ec.europa.eu/environment/ chemicals/lab_animals/home_en.htm). All animal work at Wageningen University was approved by the local experimental animal committee (DEC number 2014095).

## Decision letter and Author response
Decision letter https://doi.org/10.7554/eLife.64520.sa1
Author response https://doi.org/10.7554/eLife.64520.sa2

## Additional files
### Supplementary files
• Transparent reporting form

## Data availability
All data generated or analyzed during this study are included in the manuscript and supporting files. Source data files have been provided for Figures 1, 2, 4, 5, 6, 7, and 9.

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
