## [Decision Letter]

**Acceptance summary:**

In this study, Jacobs and colleagues use a zebrafish infection model to investigate innate immune responses to extracellular trypanosome infections. Differences in early macrophage responses in individual animals strongly influenced infection outcomes. In particular, induction of a controlled inflammatory response was associated with low parasitaemia and host survival, while failure to control infection was associated with an exacerbated inflammatory response and the appearance of foamy macrophages. Thus trypanosome infections can be controlled by innate immune responses, although the over-activation of these responses and formation of foamy macrophages exacerbates disease.

**Decision letter after peer review:**

Thank you for submitting your article "Occurrence of foamy macrophages during the innate response of zebrafish to trypanosome infections" for consideration by *eLife*. Your article has been reviewed by 3 peer reviewers, including Malcolm J McConville as the Reviewing Editor, and the evaluation has been overseen by Carla Rothlin as the Senior Editor. The following individual involved in review of your submission has agreed to reveal their identity: Carl De Trez.

The reviewers have discussed the reviews with one another and the Reviewing Editor has drafted this decision to help you prepare a revised submission.

Summary:

This study investigates the role of the innate immune response in controlling bloodstream trypanosome infection in the zebrafish infection model recently developed by the authors. The study found that an innate immune response, characterized by controlled inflammatory response was sufficient to control infection in some individuals, while failure to control infection was associated with a strong inflammatory response characterized by expansion of foamy macrophages. The findings highlight the importance of a balanced immune response in controlling bloodstream trypanosome infections that are likely relevant to mammalian infections.

Essential revisions:

1. Can the authors provide further quantification and statistical analysis of some of the interesting phenotypes reported in Figure 7, 8, 9 and 10. In particular:

– The authors report that macrophages, but not neutrophils, infiltrate in the cardinal vein, although both cell populations are accumulating on the outer side of the vasculature during infection. Also, do neutrophils ever interact with trypanosomes in other sections of the vasculature, if not in the cardinal vein? Do trypanosomes ever escape from the circulation and interact with neutrophils elsewhere?

– The authors report that foamy macrophages occur inside the vasculature and are exclusive to high-infected larvae. Can they quantitate these associations and perform statistical tests (i.e. count foamy/non-foamy mpeg+ cells inside/outside the vessels in the PVP, T. car-low and T. car-high groups)? Is there a possibility that macrophages are scavenging dead Trypanosoma from the circulation, and is this leading to the foamy macrophage phenotype? Trypanosomes are also leading to hemolysis and this could lead to increased phagocytosis of red blood cell debris by macrophages. Could this be linked to the foamy appearance? How specific is BODIPY, to distinguish cholesterol (typical of foamy macrophages), vs lipids derived by phagocytosis of cell debris (i.e. high in membrane phospholipids?)

– The authors report that foamy macrophages occurring in T. car-infected larvae are characterised by a strong pro-inflammatory profile and are all il1beta and all tnfa positive. Significant differences are observed in the inflammatory response of macrophages in high- and low-infected individuals and in their susceptibility to infection. Can the authors show that foamy macrophages are indeed more frequently il1b positive/tnfa positive than neighbouring non-foamy mpeg+ cells?

– The authors report that a strong inflammatory profile is associated with the occurrence of foamy macrophages. However, it is not clear how widely spread the inflammation is and only images of macrophages and endothelial cells in the cardinal vein are shown. Moreover, only tnfa and il1b are assessed (using transgenic reporters). The authors also mention that they observe a mild inflammatory response in low-infected individuals and that this is strongly associated with control of parasitaemia and survival to the infection. Can they confirm strong vs mild inflammatory profiles and different association with survival in the 2 infection categories and PVP control with a panel of qRT-PCR for several inflammatory markers (i.e. il1beta, tnfa and other relevant cytokines and chemokines)?

2. Figure 1 diagram is very useful, but could be enhanced by inclusion of a representative quantification in an additional panel. For example, when injecting 200 *T. carassii*, what % of larvae is classified in the two infection categories? Could the authors also further discuss the % of T. low larvae where no parasites were observed during the clinical scoring? Have these larvae (or some of them) cleared the infection completely?

3. Figure 2: Is the clinical scoring predictive of early death onset (or likelihood of death)? To show this, can the authors divide the T. car 200 survival curve into 2 separate curves, based on the clinical scoring at day 4-5.

4. Authors are discouraged from using the term macrophage/neutrophil proliferation In Figure 5 and Figure 6 and related text. Normally "macrophage proliferation" is used to refer to resident tissue macrophages that occasionally are seen to divide/proliferate. The authors are more likely detecting myelopoiesis (in agreement, Edu staining most commonly is seen in hematopoietic tissues) and the EdU staining in mature macrophages/neutrophils is the result of a (recent) cell division of a hematopoietic progenitor cell. In the absence of a more specific mechanistic insight, the use of much broader terms, such as "increased production/number of macrophages/neutrophils" rather than "macrophage/neutrophil proliferation", is encouraged.

5. The authors refer to the trypanosuceptible vs. trypanotolerant background of the host observed in non-zebrafish models. However, in this particular setting, all the larvae possess an identical genetic background. Therefore, why would the larvae behave differently in response to a similar pathogen? In addition, there is no clear differences in neither parasitic load at 2 dpi (figure 3F) nor myeloid cells accumulation at 3 dpi (figure 4AB), which could lead to a drastic difference in parasitic load based on mRNA expression at 4 dpi (figure 3F). The authors should discuss this shortly.

6. Figure 4: the representative pictures from FigB do not seem to clearly match the histograms depicted in Figure 4C. For example, from the pictures in Figure 4B, it seems that there is a decrease in red fluorescence in the representative pictures from 7 dpi to 9 dpi low-infected larvae, which is not reflected in the histogram. Also, representative picture of 7 hi-infected larvae seems to show at least equal or even more red fluorescence compared to 9 dpi low-infected larvae.

---

## [Author Response]

Essential revisions:1. Can the authors provide further quantification and statistical analysis of some of the interesting phenotypes reported in Figure 7, 8, 9 and 10. In particular:– The authors report that macrophages, but not neutrophils, infiltrate in the cardinal vein, although both cell populations are accumulating on the outer side of the vasculature during infection. Also, do neutrophils ever interact with trypanosomes in other sections of the vasculature, if not in the cardinal vein? Do trypanosomes ever escape from the circulation and interact with neutrophils elsewhere?

Trypanosomes certainly do extravasate, not only to the peritoneal cavity but also to tissues and interstitial fluids (See figure 2, Supplementary Video 1 and previous report Doro et al. 2019, *eLife;* DOI:10.7554/*eLife*.48388). Trypanosomes however are very motile parasites and, especially in compact tissues of the fins and muscle, can tumble very rapidly or swim directionally at speeds ranging between 45 and 57 µm/sec (Video 7 – 27 sec, in Doro et al. 2019, *eLife*). We believe that given the high motility of trypanosomes, either as directional swimmers as well as tumblers, it is unlikely for immune cells to phagocytose, or ‘chase’ and directly interact with trypanosomes.

Nevertheless, we realize that supplementary video 1 in the current manuscript does not provide a reference for trypanosome motility relative to neutrophil location or motility. Therefore, we now provide an additional supplementary video showing neutrophils surrounded by motile trypanosomes; this video shows how fast a single trypanosome can pass by or swim away from a neutrophil (or any other immune cell). Currently, we find it most suited to add the video in the Discussion section after the current sentence ‘Phagocytosis however, can be excluded as one of the possible contributing factors since motile *T. carassii*, similar to other extracellular trypanosomes (Caljon et al., 2018; Saeij et al., 2003; Scharsack et al., 2003), cannot be engulfed by any innate immune cell (Video 7).’ Line 630.

– The authors report that foamy macrophages occur inside the vasculature and are exclusive to high-infected larvae. Can they quantitate these associations and perform statistical tests (i.e. count foamy/non-foamy mpeg+ cells inside/outside the vessels in the PVP, T. car-low and T. car-high groups)?

Following the reviewers’ suggestions, in figure 6 we now provide a quantification of the total number of *mpx*:GFP+ (neutrophils) and *mpeg1.4*:mCherry-F+ (macrophages) inside/outside the vessels. Of the macrophages that were inside the vessels, we also quantified the proportion of those having a rounded, granular morphology and confirm that rounded and granular macrophages are exclusively found inside the vessel of high-infected individuals.

Next, in figure 7, we established that the rounded, granular macrophages found exclusively inside the vessel in figure 6 and 7A, stain positive for the BODIPY FLC5. Thus, we established that these are foamy macrophages. Following the reviewers request, we now provide a quantification of their number and confirm that these occur exclusively in high-infected individuals. In the revised figure 7, we now also provide a representative image of a lowinfected individual, showing no positivity for the BODIPY FLC5 dye in any of the macrophages present, specifying that BODIPY+ foamy macrophages only occur in high-infected individuals. Video 6 was updated accordingly.

Is there a possibility that macrophages are scavenging dead Trypanosoma from the circulation, and is this leading to the foamy macrophage phenotype? Trypanosomes are also leading to hemolysis and this could lead to increased phagocytosis of red blood cell debris by macrophages. Could this be linked to the foamy appearance? How specific is BODIPY, to distinguish cholesterol (typical of foamy macrophages), vs lipids derived by phagocytosis of cell debris (i.e. high in membrane phospholipids?)

Both hypotheses suggested by the reviewers are possible and we believe both can contribute/have contributed to the foamy macrophage phenotype. However, a third option that may explain the foamy appearance, which does not exclude the previous two and may rather be a consequence of those, is a change in macrophage lipid metabolism in *response* to phagocytosis of trypanosome-derived PAMPs and/or host DAMPs.

As also referred to later by the reviewers (comment 14), foam cells [macrophages with lipid droplets (LDs) which are stores of triacylglycerols (TAGs) AND/OR cholesterol esters (CEs)], are found in various disease states. CE-rich foam cells are typical of atherosclerotic plaques whereas TAG-rich foam cells are most commonly found in tuberculi formed during *Mycobacterium tuberculosis* infections and were described also during *T. cruzi* and Leishmania infections (Guerrini et al. 2019. doi:10.1016/j.it.2019.10.002; Saka HA and Valdivia R. 2012. doi:10.1146/annurev-cellbio-092910-153958). Thus both, CE-rich and TAG-rich foam cells exist and represent two different paradigms of foam cell formation during disease, although mixed phenotypes have also been observed.

Furthermore, most if not all studies investigating foam cells, labelled the lipid droplets in in vitro cell cultures of foamy macrophages or in fixed tissues isolated from individuals post-mortem. Most of these studies used either the neutral lipid dyes Nile Red or BODIPY 493/503, which require short incubation time but are NOT specific for foamy macrophages as they bind to anything which has neutral lipids (cholesterol) in/on it. BODIPY-FLCx (where x indicates the number of carbon atoms in fatty acids moieties of various lengths) can directly bind to lipids but can also be actively incorporated into phospholipids membranes and triacylglycerol (TAG) and can therefore be used to also investigate changes in lipid metabolism within live cells, but requires different kinetics of uptake.

In our case, wanting to investigate the formation and occurrence of foamy macrophages in vivo, during an ongoing infection in a live developing embryo, we had to consider the presence of several lipid-rich (digestive) organs (intestine, pancreas, liver, gall bladder, etc) in which lipid accumulation occurs (Carten JD, et al. 2011. doi:10.1016/j.ydbio.2011.09.010), as well as the presence of soluble lipid/cholesterol-binding proteins in the serum. As suggested by the reviewers, we also considered that foamy macrophages could be scavenging dead trypanosomes and damaged red blood cells, but we considered these stimuli triggers of a metabolic change leading to the foamy macrophage phenotype and did not consider these stimuli binders of the lipid dye for the reasons explained below.

Thus, in our study we considered both commonly used dyes, BODIPY 493/503 (labelling mostly CE) and BODIPY FLC5 (actively incorporated in TGA) and allowed them to accumulate in lipid rich cells for 20h. As expected, both dyes labelled lipid-rich organs (as previously shown in zebrafish, Carten JD, et al. 2011. doi:10.1016/j.ydbio.2011.09.010), and both were able to label the large, granular macrophages observed in the vessel of high-infected individuals (see figure 8 in manuscript and Author response image 1). Thus, we used these two labels to confirm that these cells are indeed foamy macrophages rich not only in TGA but also cholesterol. Nevertheless, a higher background was observed when using the neutral-lipid dye BODIPY 493/503, especially in serum of both, infected and non-infected individuals. The background staining was most likely due to the abundance of soluble neutral-lipids (including cholesterol) in the serum of young larvae to which the dye could directly bind. This background staining would not be seen when working with in vitro cell cultures, or with fixed tissues.

In Author response image 1 we provide images of a control (PVP) and of a high-infected individual obtained after administration of BODIPY 493/503. In the vessel of the infected individual, accumulation of the dye in foamy macrophages is clearly visible (yellow arrowheads). In the vessel of non-infected individuals, BODIPY 493/503-negative leukocytes rolling on the vessel wall (blue arrowheads) are clearly visible, confirming the absence of staining of other cells residing in the blood stream and thus the specificity of staining for foamy macrophages.

**Author response image 1. sa2fig1:** 

For the reasons mentioned above, and since BODIPY FLC5 can be incorporated into de novo TAG, dead parasites or damaged red blood cells that may have been phagocytosed may not contribute to de novo TGA synthesis and thus to the labelling in macrophages. Furthermore, if the dye would specifically bind to cell membranes (phospholipids), at 20h after BODIPY administration we should have observed a much higher background labelling in parasites or red blood cells circulating in the blood stream, or in parasites or cells surrounding the blood vessels, that was the case.In the revised figure 7 it is clearly visible that, except for endothelial cells, which are positively labelled by the Bodipy in all groups, only foamy macrophages in high-infected individuals are positive for the BODIPY FLC5, and none of the neighbouring (non-foamy) macrophages in the same individual, or in PVP or low-infected individuals, are positive for the dye. These observations now have been quantified in Figure 7C.

For all of the above reasons, we prefer to show in the paper the data obtained using the BODIPY FLC5 dye.

– The authors report that foamy macrophages occurring in T. car-infected larvae are characterised by a strong pro-inflammatory profile and are all il1beta and all tnfa positive. Significant differences are observed in the inflammatory response of macrophages in high- and low-infected individuals and in their susceptibility to infection. Can the authors show that foamy macrophages are indeed more frequently il1b positive/tnfa positive than neighbouring non-foamy mpeg+ cells?

As requested by the reviewers, in figure 9C of the revised manuscript, we now quantified the number of foamy and non-foamy macrophages being positive for *Il1b* or *tnfa*. The new data confirm that 100% of the foamy macrophages is positive for *Il1b* or *tnfa* and that high-infected individuals generally have a higher number of *il1b* or *tnfa* positive macrophages then low-infected or PVP individuals.

– The authors report that a strong inflammatory profile is associated with the occurrence of foamy macrophages. However, it is not clear how widely spread the inflammation is and only images of macrophages and endothelial cells in the cardinal vein are shown. Moreover, only tnfa and il1b are assessed (using transgenic reporters). The authors also mention that they observe a mild inflammatory response in low-infected individuals and that this is strongly associated with control of parasitaemia and survival to the infection. Can they confirm strong vs mild inflammatory profiles and different association with survival in the 2 infection categories and PVP control with a panel of qRT-PCR for several inflammatory markers (i.e. il1beta, tnfa and other relevant cytokines and chemokines)?

At the start of our study, prior to the provision of the *tnfa* and *il1b* transgenic lines, we had performed the gene expression analysis also suggested by the reviewers. However, based on the obtained results and the overall pattern of expression of *tnfa:gfpF* and *il1b:gfpF* in zebrafish larvae obtained later, we realized that measuring gene expression in *whole larvae* at this developmental stage may not be very sensitive in revealing (subtle) differences in inflammatory responses during infection. In fact, in fish as in mammals, expression of most immune genes of interest is not specific to immune cells only. For example, when the *il1b:gfpF* transgenic line was established by our collaborators (Nguyen-Chi et al. 2014, *DMM*; doi:10.1242/dmm.014498) it was shown that although GFP expression was highly inducible in cells of myeloid origin (macrophages and neutrophils), a generalized basal expression of *il1b:gfp* was detected in skin keratinocytes, neuromasts and gut enterocytes, all cell types in direct contact with the environment. Those data were supported by in situ hybridization and also by comparison with the ‘spontaneous expression of GFP in the Tg(NFκB:EGFP) reporter transgenic zebrafish line (Kanther and Rawls, 2010), which is in agreement with the well-known contribution of NF-κB to il1b induction (Ogryzko et al., 2014)’. Similar observations were made for the *tnfa:gfpF* line. Furthermore, in our study, we noticed that larvae older than 5 dpf (thus older than in the original study), show *tnfa:gfpF* and *il1b:gfpF* expression also in the developing haematopoietic organs (head kidney and spleen in the head region) and in endothelial cells of the caudal vein.

Taking all this into account we propose to provide the gene expression analysis data as supplementary figure (Figure 2-supplement 1) with the accompanying text. Lines 219-229.

*“*A preliminary gene expression analysis of a panel of immune-related genes was performed on pools of larvae classified as high- or low-infected. […] For these reasons, taking advantage of the transparency of zebrafish larvae and of the established clinical scoring system, subsequent characterization of the inflammatory response to *T. carassii* infection, was performed on individual larvae, focusing on innate immune cells.”

To better address the first part of the question of the reviewers (“how widely spread the inflammation is”) we first would like to remind the reviewers that, besides the head region (where haematopoiesis takes place in the developing larvae), the majority of leukocytes in larval zebrafish are concentrated in the trunk and tail region, distributed along the two major blood vessels (dorsal aorta/caudal artery and posterior cardinal vein/caudal vein) including the tail tip loop. See also figure 4 in the manuscript and the examples provided in Author response image 2 (4 dpi embryo, macrophages (red) and neutrophils (green)).Also relevant is that, for the images used in the main paper, to be able to reliably count macrophages and neutrophils, or to identify cytokine-expressing macrophages and their specific location (in/outside vessels), we had to work with confocal images acquired at 40x magnification, which explains why often only a small section of the caudal vein is shown.

We certainly agree that it is interesting to show how widely spread the inflammation is. To this end, in the revised manuscript we now provide an additional supplementary figure (S3 Fig) showing the widespread distribution of *tnfa*:GFP positive (*tnfa*:GFP+) cells in four subsequent locations, covering the entire trunk and tail region. As expected, *tnfa*:GFP+ leukocytes are distributed mostly along the major vessels in the trunk region, and along the tail tip loop and fin of the tail region. The images also show that not only the number of *tnfa*:GFP+ cells but also the intensity of their GFP signal is higher in high-infected individuals, compared to PVP or low-infected individuals, confirming a widely spread and higher inflammatory state in high-infected individuals. It may also be clear from these images that, as mentioned earlier, in zebrafish as in mammals, *tnfa* expression is not always exclusive to immune cells only. The widespread expression of *tnfa* in non-immune cells can have a confounding effect when it comes to measuring gene expression in whole larvae, we therefore think that (also) showing these images can be more informative than showing (only) gene expression. Imaging allows us to maximally take advantage of the zebrafish model to *visualize* and *describe* responses where they occur.Furthermore, based on the comments of the reviewers we realised that we indeed drew attention to the caudal vein, whereas the caudal vein is NOT the only vessel shown in our images. The dorsal aorta (DA) or caudal artery (CA) are right above the posterior cardinal vein (PCV) and caudal vein (CV), respectively and, depending on the orientation of the embryo and specific focus or zoom level shown, all are often visible in our images or in the supplementary videos. In the revised manuscript, whenever possible, and if it did not interfere with the clarity of the image (e.g Figure 2, 6, 7A etc.), we now also indicate the presence of the DA or CA, and when relevant, also refer to these vessels in the main text.

2. Figure 1 diagram is very useful, but could be enhanced by inclusion of a representative quantification in an additional panel. For example, when injecting 200 T. carassii, what % of larvae is classified in the two infection categories? Could the authors also further discuss the % of T. low larvae where no parasites were observed during the clinical scoring? Have these larvae (or some of them) cleared the infection completely?

In response to the reviewer’s suggestion, we have added a new graph to figure 2 (Figure 2F) showing the proportion of larvae allocated to each clinical score of a total of 350 larvae. In the corresponding Results section, we now mention that larvae with a score equal to 1, in which no parasites were observed using the second criterion (ratio of parasite:RBC in intersegmental capillary (ISC)), where still found to be infected when using the third criterion, extravasation. Thus, although parasites could not be seen in the ISC, they already extravasated to the tissue, indicating that at this point after infection larvae cannot clear the infection. We have further clarified this in the methods section of the revised manuscript and in the corresponding Results section of figure 2.

3. Figure 2: Is the clinical scoring predictive of early death onset (or likelihood of death)? To show this, can the authors divide the T. car 200 survival curve into 2 separate curves, based on the clinical scoring at day 4-5.

In response to the reviewers’ suggestion, we have added a new graph to figure 2 (Figure 2G) showing the survival rate of the control (PVP), *T. carassii*-low and -high groups (50 larvae per group). All high-infected individuals succumbed to the infection, whereas low-infected individuals had a high relative percent survival (82%) when compared to the PVP group. Of the low-infected individuals that were removed from the experiment, none displayed high parasitaemia, indicating that they did not succumb due to the infection. Instead, they showed developmental delays such as block in blood circulation, stunt growth, etc., similar to those observed in non-infected control individuals in the PVP group.

4. Authors are discouraged from using the term macrophage/neutrophil proliferation In Figure 5 and Figure 6 and related text. Normally "macrophage proliferation" is used to refer to resident tissue macrophages that occasionally are seen to divide/proliferate. The authors are more likely detecting myelopoiesis (in agreement, Edu staining most commonly is seen in hematopoietic tissues) and the EdU staining in mature macrophages/neutrophils is the result of a (recent) cell division of a hematopoietic progenitor cell. In the absence of a more specific mechanistic insight, the use of much broader terms, such as "increased production/number of macrophages/neutrophils" rather than "macrophage/neutrophil proliferation", is encouraged.

Thank you for the suggestion, we have now amended the text and either refer to EdU+ or to (recently) divided neutrophils/macrophages.

5. The authors refer to the trypanosuceptible vs. trypanotolerant background of the host observed in non-zebrafish models. However, in this particular setting, all the larvae possess an identical genetic background. Therefore, why would the larvae behave differently in response to a similar pathogen? In addition, there is no clear differences in neither parasitic load at 2 dpi (figure 3F) nor myeloid cells accumulation at 3 dpi (figure 4AB), which could lead to a drastic difference in parasitic load based on mRNA expression at 4 dpi (figure 3F). The authors should discuss this shortly.

We are not sure whether we fully understand this comment of the reviewer, therefore we first would like to clarify the following points. Trypanosusceptible and trypanotolerant mice are inbred and therefore the response to trypanosomes infection will be very similar among individuals within each of these population/strains, but will be contrasted between the two mouse strains. Our zebrafish have indeed a similar genetic background because they come from the same parental line but they are *not* identical, therefore individual differences can and will occur in response to infections, as observed in the current study. Although the parasite load and myeloid cell number did not differ at 2 and 3 dpi, respectively, other factors can be different between individuals, likely due to genetic differences, and may account for part of the observed variation.

It was not our intention to draw attention to the genetic background of mice or zebrafish, but rather to the fact that by using transparent animals we were able to detect differences in response at the individual level. The contrasting responses in high- and low-infected zebrafish may resemble the contrasting responses in trypanosusceptible and trypanotolerant mice. We merely aimed to address the similarity in *response* rather than address genetic influences. To not confuse the future readership, we therefore removed the specific reference to trypanosusceptible and trypanotolerant mice and simply refer to the studies that have used these mice strains to elucidate the type of response considered beneficial or detrimental to the infection. Line 646-648.

6. Figure 4: the representative pictures from FigB do not seem to clearly match the histograms depicted in Figure 4C. For example, from the pictures in Figure 4B, it seems that there is a decrease in red fluorescence in the representative pictures from 7 dpi to 9 dpi low-infected larvae, which is not reflected in the histogram. Also, representative picture of 7 hi-infected larvae seems to show at least equal or even more red fluorescence compared to 9 dpi low-infected larvae.

We thank the reviewers for such precise observation. We have now updated the representative images in the figure 3B.